# ONLINE INVENTORY OPTIMIZATION IN NON-STATIONARY ENVIRONMENT

**Koji Ichikawa, Kei Takemura & Tatsuya Matsuoka**
NEC Corporation
Tokyo, Japan
{k_ichikawa0, kei_takemura, ta.matsuoka}@nec.com

## ABSTRACT

This paper addresses online inventory optimization (OIO), an extension of online convex optimization. OIO is a sequential decision-making process in inventory management cycles consisting of order arrival, stock consumption, and new order placement. One key challenge in OIO is managing demand fluctuations. However, most existing algorithms still cannot sufficiently handle this because they focus on a static regret guarantee, comparing their performance to a fixed order-up-to level strategy. In non-stationary environments, such a static comparator is unsuitable due to demand fluctuations. In this paper, we propose an algorithm with near-optimal dynamic regret guarantee for OIO. Our algorithm also offers an improvement of $\sqrt{L_{\max}}$ for the static regret upper bound in existing studies. Here, $L_{\max}$ refers to the maximum sell-out period. Our algorithm employs a simple two-stage projection strategy, through which we prove that the OIO is connected to the smoothed online convex optimization.

## 1 INTRODUCTION

Inventory management is crucial in supply chain management, with extensive research focusing on optimal ordering strategies for various inventory systems. In particular, systems with periodic reviews and carryover stock are closely related to real-world problems. Numerous approaches have been proposed for these systems, assuming known demand models (see, e.g., Glock et al. (2014)). However, it is often challenging to obtain a complete demand model in advance, which highlights the necessity for online learning techniques to adapt to unknown demands.

Recently, Online Convex Optimization (OCO) (Hazan, 2016; Orabona, 2019; Shalev-Shwartz, 2012) has attracted attention in the online inventory management (Huh & Rusmevichientong, 2009; Shi et al., 2016; Zhang et al., 2018a; 2020; Yuan et al., 2021; Agrawal & Jia, 2022; Hihat et al., 2023). OCO is a sequential learning framework in which for each round $t \in [T]$, the decision maker chooses an $N$-dimensional vector $y_t$ that is in a convex feasible region $\mathcal{C} \subset \mathbb{R}^N$ and then the environment reveals a convex loss function $\ell_t$. The typical aim of the decision maker is to minimize the static regret, $\sum_t \ell_t(y_t) - \min_{u \in \mathcal{C}} \sum_t \ell_t(u)$. Here we note that in the inventory system, most loss functions, such as the Newsvendor loss, are convex.

Online inventory optimization (OIO) is a variant of OCO, formulated by Hihat et al. (2023). In OIO, a sequential decision-making process involving the inventory cycle of order arrival, stock consumption, and new order placement is considered. During each round $t \in [T]$, the stock is replenished to the order-up-to level of $y_t$ set in the previous round. The environment processes the subsequent demand and post-processing activities, revealing an $N$-dimensional carryover stock level of $x_{t+1}$ and a subgradient $g_t \in \partial \ell_t(y_t)$ that is associated with the convex loss incurred by the decision $y_t$. Then, the decision maker determines the next order-up-to level $y_{t+1}$ that is greater than $x_{t+1}$ and less than the capacity constraint of the warehouse. In the OIO setting, Hihat et al. (2023) have proposed the MaxCOSD algorithm, which achieves a sublinear static regret.

However, the static regret guarantee is not sufficient for practical applications, especially in environments with demand fluctuations. Consider a simple example of a single-item inventory system with a capacity limit of $D$. Set the fluctuating demand as $d_t = Dt/T$ for $t \in [T]$ and the loss function as the Newsvendor loss of $\ell_t(y) = |y - d_t|$. A straightforward calculation shows that the minimum

total loss of the static comparator is $\min_{u \in [0,D]} \sum_{t=1}^{T} \ell_t(u) = \mathcal{O}(DT)$, whereas a time-varying comparator with $u_t = d_t$ results in $\sum_{t=1}^{T} \ell_t(u_t) = 0$. Thus, even if we have an algorithm with $\mathcal{O}(\sqrt{T})$-static regret for this example, it may still suffer from $\Omega(T)$-regret when comparing it to $u_t$.

Recent studies on OCO have intensively investigated algorithms for dynamic environments (Hall & Willett, 2013; Zhang et al., 2018b; Zhao et al., 2020; 2024). The dynamic regret is an indicator that measures an algorithm's tolerance against changing environments. In the context of OCO, the dynamic regret is defined as $R_T(u_1, \ldots, u_T) := \sum_{t=1}^{T} \ell_t(y_t) - \sum_{t=1}^{T} \ell_t(u_t)$, which is a function of a time-varying comparator sequence $u_1, \ldots, u_T$. A major approach for the dynamic regret is based on a two-layer structure, where a meta-algorithm adaptively accumulates the decisions of a set of base learners (Zhang et al., 2018b; van Erven et al., 2021; Zhang et al., 2022b). Such algorithms ensure $\mathcal{O}(\sqrt{(1 + P_T)T})$-dynamic regret, where $P_T$ is the total path-length of the comparator: $P_T := \sum_{t=2}^{T} \|u_{t-1} - u_t\|_1$. [1] Therefore, in OIO, a key question is *whether we can construct an algorithm that ensures an $\mathcal{O}(\sqrt{(1 + P_T)T})$-dynamic regret in the OIO setting*. If we have such an algorithm, we obtain a sublinear dynamic regret for the aforementioned example because $P_T = \sum_{t=2}^{T} D/T = \mathcal{O}(D)$.

One major difficulty in the dynamic regret minimization for OIO is the carryover stock constraint. While the order-up-to level $y_t$ must be greater than the carryover stock $x_t$, the comparator $u_t$ is not subject to this constraint. Thus, the feasible region of $u_t$ is always a superset of that of $y_t$. Most algorithms for OCO provide regret guarantees only for comparators $\hat{u}_t$ that are in the same feasible region as $y_t$. Consequently, this naive application results in $\mathcal{O}(T)$-regret due to the gap between $\hat{u}_t$ and $u_t$. For the static regret minimization, Hihat et al. (2023) overcome this difficulty by a cyclical update approach, where $y_t$ is only updated to a candidate $\hat{y}_t$ when $\hat{y}_t$ is feasible.

When considering the dynamic regret, however, we cannot employ a standard two-layer structure with an OIO algorithm (such as MaxCOSD) as the base learner to leverage its theoretical guarantees. A fundamental difficulty is that this architecture contradicts a key assumption for OIO algorithms: the carryover stock level $x_{t+1}$ must be less than the preceding replenished stock level $y_t$. A meta-algorithm's decision $y_t$ might be larger than the output $y_t^a$ of a base learner $a$. With a small demand, $x_{t+1}$ can exceed $y_t^a$. For the base learners, this carryover stock level violates their assumption ($x_t^i \leq y_t^{ai}$ for all $i$). This inconsistency prevents us from obtaining a theoretical guarantee for the two-layer structure.

## 1.1 CONTRIBUTIONS

The main contribution of this paper is to propose OIO algorithms with near-optimal dynamic regret guarantee, as stated in the following theorem.

**Theorem 1** (Informal). *Under the constraints of carryover stock and the warehouse capacity, there exists an algorithm that ensures*

$$R_T(u_1, \ldots, u_T) \leq \tilde{\mathcal{O}}\left(\sqrt{L_{\max} T (1 + P_T)}\right),$$

*for any comparator sequence $u_1, \ldots, u_T$, without knowing $L_{\max}$ and $P_T$ a priori.*

Here, $\tilde{\mathcal{O}}$ is an order symbol that ignores logarithmic factors. $L_{\max}$ is the maximum sell-out period defined in Definition 1, which, informally speaking, indicates that the total demand over $L_{\max}$ rounds is at least the warehouse capacity. For static regret, the algorithm guarantees $\tilde{\mathcal{O}}(\sqrt{L_{\max} T})$-regret, offering an improvement of $\sqrt{L_{\max}}$ over the existing works. The regret bounds are summarized in Table 1.[2]

Our algorithm employs a simple two-stage projection strategy consisting of a base learner and its projection onto a feasible region. In each round $t$, an observed subgradient $g_t$ is fed to the base

---

[1] We note that, in the standard OCO setting, a simpler algorithm such as Online Gradient Descent algorithm (OGD) can also achieve $\mathcal{O}(\sqrt{(1 + P_T)T})$-dynamic regret upper bound if the learner knows $P_T$ in advance. However, this parameter is generally difficult to set *a priori*, which motivates the use of a meta-algorithm.

[2] The parameters corresponding to $L_{\max}$ in each reference are as follows: $1/\gamma$ in Huh & Rusmevichientong (2009); $1/\mu$ in Zhang et al. (2018a); $1/c_2$ in Zhang et al. (2020); $D$ in Agrawal & Jia (2022); $\rho\beta$ in Yuan et al. (2021); $1/l$ in Shi et al. (2016); and $1/\mu$ in Hihat et al. (2023).

Table 1: Regret bounds of [1] Huh & Rusmevichientong (2009); [2] Zhang et al. (2018a); [3] Zhang et al. (2020); [4] Agrawal & Jia (2022); [5] Yuan et al. (2021); [6] Shi et al. (2016); [7] Hihat et al. (2023); and our work. In the table, we list the regret bounds for each reference by replacing the demand characteristic parameters used in each paper with our indicator $L_{\max}$. $C$ in the fifth row is a positive constant which depends on other parameters. In the references marked with a dagger, lead time is taken into account. We show the regret bounds when the lead time is equal to one. S/M in the Item column represents single-/multi-item setting. NV, O, and F in the Loss column represent Newsvendor loss, outdating cost, and fixed cost, respectively. The warehouse capacity constraint used in each reference is given in the Capacity column: Interval ($y_t \in [0, D]$, where $D$ denotes the capacity); Linear ($\sum_i y_t^i \leq D$); and Convex ($y_t \in \mathcal{C}$ for a convex set $\mathcal{C}$).

| Regret | Reference | Upper Bound | Lower Bound | Item | Loss | Demand | Capacity |
|--------|-----------|-------------|-------------|------|------|--------|----------|
| Static | [1] | $\mathcal{O}(L_{\max}\sqrt{T})$ | | S | NV | i.i.d. | Interval |
| | [2] | $\mathcal{O}(L_{\max}\sqrt{T})$ | $\Omega(\sqrt{T})$ | S | NV + O | i.i.d. | Interval |
| | [3]$^\dagger$ | $\mathcal{O}(L_{\max}\sqrt{T})$ | $\Omega(\sqrt{T})$ | S | NV | i.i.d. | Interval |
| | [4]$^\dagger$ | $\tilde{\mathcal{O}}(\sqrt{T} + L_{\max})$ | | S | NV | i.i.d. | Interval |
| | [5] | $\tilde{\mathcal{O}}(e^{CL_{\max}}\sqrt{T})$ | | S | NV + F | i.i.d. | Interval |
| | [6] | $\mathcal{O}(L_{\max}\sqrt{T})$ | | M | NV | indep. | Linear |
| | [7] | $\mathcal{O}(L_{\max}\sqrt{T})$ | | M | Convex | non-i.i.d. | Convex |
| Static | **[This work]** | $\tilde{\mathcal{O}}(\sqrt{L_{\max}T})$ | $\Omega(\sqrt{L_{\max}T})$ | M | Convex | non-i.i.d. | Linear |
| Dynamic | | $\tilde{\mathcal{O}}(\sqrt{L_{\max}(1 + P_T)T})$ | | | | | |

learner to propose a decision $\hat{y}_{t+1}$, which is then adjusted to $y_{t+1}$ to meet carryover stock constraints. A distinctive feature of our algorithm is that the base learner's decision is made independently of the carryover stock. We note that our update process differs from MaxCOSD's in that ours allows the order-up-to level $y_t$ to change, even if the base learner's decision $\hat{y}_t$ is infeasible.

Our primary technical contributions are twofold. First, we demonstrate that, under our two-stage projection, the dynamic regret can be bounded by the base learner's regret with switching costs proportional to $L_{\max}$, which eliminates the concerns regarding the dynamic carryover stock constraint. Leveraging this result, we achieve a near-optimal dynamic regret by employing an algorithm for well-known Smoothed OCO (SOCO) (Lin et al., 2011; Zhang et al., 2021; 2022c;a) as the base learner, along with the doubling trick for unknown $L_{\max}$. Second, we provide, for the first time, a $\Omega(\sqrt{L_{\max}T})$ lower bound for the OIO setting. Our matching upper and lower bounds establish that $\tilde{\mathcal{O}}(\sqrt{L_{\max}T})$ is nearly optimal, which resolves the open question raised by Hihat et al. (2023).

## 2 RELATED WORKS

**Inventory Management** Inventory management is a long-standing research topic in the field of operations research. It addresses various conditions, such as demand model (deterministic or stochastic), carryover status (stateless or stateful), review frequencies (periodic or continuous), lead times (constant or probabilistic), item types (single or multiple), stockout types (backorders or lost opportunities), ordering costs (linear or non-linear, with or without fixed order cost), disposal losses, multi-echelon systems, and more (see, e.g., Zipkin (2000); Porteus (2002)). In particular, a stateful inventory system with periodic reviews, i.e., a situation where the remaining stock from the previous period is carried over, is closely related to real-world problems. Numerous methods have been proposed for scenarios where the demand model is known in advance (Glock et al., 2014). However, in many cases, obtaining a complete demand model in advance is challenging. This difficulty highlights the importance of online learning for inventory optimization. As the objective function is often convex (e.g., the Newsvendor loss), various studies have explored this online inventory optimization problem in relation to OCO problems (Huh & Rusmevichientong, 2009; Shi et al., 2016; Zhang et al., 2018a; 2020; Yuan et al., 2021; Agrawal & Jia, 2022; Hihat et al., 2023).

**Online Convex Optimization** Online convex optimization (OCO) (Shalev-Shwartz, 2012; Hazan, 2016; Orabona, 2019) is a sequential learning framework that chooses $y_t$ and minimizes regret $\sum_t \ell_t(y_t) - \sum_t \ell_t(u)$ for a convex time-varying function $\ell_t$. It is shown that Online Gradient Descent algorithm (OGD) achieves the minimax optimal regret bound of $\mathcal{O}(\sqrt{T})$ (Zinkevich, 2003; Abernethy et al., 2008). For an exp-concave loss function, faster convergence can be achieved by Online Newton Step algorithm (Hazan et al., 2007), which enjoys a static regret bound of $\mathcal{O}(\log T)$.

In OCO, one of the important topics is developing algorithms that adapt to dynamic environments. There are two major performance metrics: dynamic regret and (strongly) adaptive regret. Dynamic regret, also known as switching or tracking regret, is defined as $R_T(u_1, \ldots, u_T) := \sum_{t=1}^{T} \ell_t(y_t) - \sum_{t=1}^{T} \ell_t(u_t)$ (Hall & Willett, 2013; Zhang et al., 2018b; Zhao et al., 2020; 2024). In Zhang et al. (2018b), it is shown that a two-layer algorithm called Ader achieves the optimal regret upper bound of $\mathcal{O}(\sqrt{(1+P_T)T})$. Adaptive regret (also known as interval regret) is defined as $R_T([s, e]) := \sum_{t \in [s,e]} \ell_t(y_t) - \min_{u \in \mathcal{C}} \sum_{t \in [s,e]} \ell_t(u)$. Here the regret is a function of the interval $[s, e] := s, s + 1, \ldots, e - 1, e$, where $1 \le s \le e \le T$. A weaker definition, considering the maximum regret, has been first proposed by Hazan & Seshadri (2007). Later on, Daniely et al. (2015) have extended it to account for any interval length. Jun et al. (2017) have proposed an algorithm achieving an adaptive regret of $\mathcal{O}(\sqrt{\tau \log T})$, where $\tau$ represents the length of the interval considered.

Smoothed OCO (SOCO) is a variant of OCO that incorporates the switching cost $\lambda \|y_t - y_{t+1}\|$ into the regret. The concept of switching cost is first motivated by data center management (Lin et al., 2011) and in the standard setting, the cost function $\ell_t$ is provided before making the decision $y_t$ (Bansal et al., 2015; Chen et al., 2018; Goel & Wierman, 2019; Goel et al., 2019). In the setting where the decision is made before observing the loss, OGD can achieve $\mathcal{O}(\sqrt{\lambda T})$ static regret (see, for example, Zhang et al. (2022a)). Zhang et al. (2021) have proposed an algorithm for the dynamic regret minimization based on Ader algorithm (Zhang et al., 2018b). Besides, it is pointed out that algorithms for OCO with memory guarantee the adaptive regret for SOCO (Zhang et al., 2022c; Gradu et al., 2023). Recently, Zhang et al. (2022a) have proposed an algorithm that guarantees upper bounds for both dynamic and adaptive regret by utilizing Discounted-Normal-Predictor (Kapralov & Panigrahy, 2011).

## 3 PROBLEM SETTING

---
**Algorithm 1** Setting of online inventory optimization
---
1: Initialize the inventory level $x_1 \in \mathcal{C}(\mathbf{0})$, order-up-to level $y_1 \in \mathcal{C}(x_1)$, where $\mathcal{C}$ is defined in Eq. (4).
2: **for** $t = 1, \ldots, T$ **do**
3:     Observe an inventory level $x_{t+1}$ that satisfies $x_{t+1}^i \in [0, y_t^i]$ for all $i \in [N]$.
4:     Observe a subgradient $g_t \in \partial \ell_t(y_t)$.
5:     Decide the next order-up-to level $y_{t+1}$ that satisfies $y_{t+1} \in \mathcal{C}(x_{t+1})$.
6: **end for**

---

We consider the online inventory optimization problem for $N$ items. The stock levels of each item are represented by components of a $N$-dimensional vector, which is an element of a convex set $\mathcal{C} \subset \mathbb{R}_{\ge 0}^N$ that defines the capacity constraints of the warehouse. At each round $t \in [T]$, the decision maker receives the order placed in the previous round, resulting in the stock level reaching the order-up-to level $y_t$. Following this, the environment processes the subsequent demand and necessary post-processing activities, revealing a carryover stock level of $x_{t+1}$ to the decision maker. Concurrently, a subgradient $g_t \in \partial \ell_t(y_t)$ that is associated with the convex loss incurred by the decision $y_t$ is observed. Then, the decision maker determines the next order-up-to level $y_{t+1}$ such that $y_{t+1} \in \mathcal{C}$ and $y_{t+1}^i \ge x_{t+1}^i$ for all $i \in [N]$. The process is summarized in Alg. 1.

**Remark 1.** *It can sometimes be challenging to observe opportunity loss. For instance, in retail stores, when an item is out of stock, customers rarely inquire with the store staff about its availability. As a result, retailers have limited knowledge about the actual demand for out-of-stock items. Recently, Hihat et al. (2023) have addressed this issue in their OIO setting, highlighting that the subgradient of the loss function can often be derived without observing the demand quantity. This is*

*because the penalty associated with the opportunity loss is typically given by multiplying the quantity of opportunity loss by a cost coefficient, as is the case with the Newsvendor loss: $p \max(0, d_t - y_t)$, where $p$ is a cost coefficient and $d_t$ and $y_t$ are demand and order-up-to level of round $t$, respectively. Since this penalty is linear with $y_t$, we can compute the subgradient only by observing whether a stockout occurred without knowing the demand quantity. Our problem setting also uses this framework.*

We consider the following three conditions. First, we consider that the replenished stock up to $y_t$ is always greater than the carryover stock level $x_{t+1}$ after subsequent demand and post-processing:

$$x_{t+1}^i = \max(0, y_t^i - d_t^i) \leq y_t^i,\tag{1}$$

for all $i \in [N]$. Here we define the demand for item $i$ at round $t$ as $d_t^i \in [0, D]$, noting that it may also include consumption from some post-processing activities.[3]

Secondly, we define the feasible region for the order-up-to level $y_t$ as the intersection of the lower bounds set by the carryover stocks

$$y_t^i \geq x_t^i \quad \forall i \in [N],\tag{2}$$

and the linear-sum constraints arising from inventory space

$$\sum_{i \in [N]} y_t^i \leq D.\tag{3}$$

Specifically, we define the function for the feasible region $\mathcal{C} : [0, D]^N \to \mathcal{P}([0, D]^N)$ as

$$\mathcal{C}(x) := \{y \in [0, D]^N \mid y^i \geq x^i \quad \forall i \in [N], \sum_{i \in [N]} y^i \leq D\}.\tag{4}$$

Finally, we assume that the subgradients of the losses are bounded:

$$\|g_t\|_2 \leq G.\tag{5}$$

In our analysis, we deal with $\ell_1$-norm of the subgradient, which is bounded as $\|g_t\|_1 \leq \sqrt{N}\|g_t\|_2 \leq \sqrt{N}G$.

We consider the adversarial environment. After observing $y_t$, the environment can choose the demand $d_t$ and convex loss function adversarially. The aim of this paper is to construct a near-optimal algorithm for OIO under the adversarial environment.

**Remark 2.** *Our study and Hihat et al. (2023) share the same setup except for the warehouse capacity constraint. While Hihat et al. (2023) assumes a general convex constraint, our work specifically addresses a linear constraint. Although the linear constraint is a special case of the convex constraint, it is commonly encountered in practical scenarios. Importantly, to our knowledge, no existing work establishes theoretically guaranteed algorithms for dynamic environments, even under the linear constraint. We also note that the linear constraint is applicable to cases in which each item occupies a different amount of space. Specifically, the weighted-sum capacity constraint, $\sum_i a^i y_t^i \leq D$ where $a^i > 0$ can be reduced to Eq. (3) by redefining $a^i y_t^i \to y_t^i$ and $a^i d_t^i \to d_t^i$.*

### 3.1 ENVIRONMENTAL DIFFICULTY INDICATOR

Algorithm's performance relies on the behavior of $x_{t+1}$, which reflects the demand and post-processing in round $t$. In our analysis, we focus on the period during which the inventory can meet demand, which is referred to as *sell-out period*.

**Definition 1** (Sell-out period). We define $L_{\max}$ as the minimum length of a period such that, for any item $i$ and any starting time $t$, the cumulative demand of item $i$ reaches $D$, the (per-item) upper bound implied by the warehouse capacity constraint.

$$L_{\max} := \min \left\{ L \in [T] \mid \sum_{s=t}^{\min(t+L-1, T+1)} d_s^i \geq D, \text{ for all } t \in [T] \text{ and } i \in [N] \right\}.$$

Here, we hypothetically assume that $d_{T+1}^i = D$.

---

[3]Even when the demand $d_t^i$ exceeds the warehouse capacity, both our algorithm and analysis remain applicable as long as the subgradient for $d_t^i > D$ is well-defined and observable, because our algorithm uses only the subgradient for updates. For example, in the Newsvendor loss, this subgradient is the same as the one observed when a stockout occurs.

Informally speaking, $L_{\max}$ indicates that each item sells at least $D$ units for any interval $[t, t + L_{\max} - 1]$. For example, for a static demand $d^i$, $L_{\max}$ is given by $\lceil D/d^i \rceil$.

We here note the relationship between the sell-out period and demand. Setting $L_{\max} = o(T)$ mildly constrains the duration of periods with small demand; this constraint prevents situations where the decision maker is forced to incur holding costs over an extended period due to the small demands. In fact, as we will show in our lower bound analysis, sublinear regret cannot be achieved when $L_{\max} = \Omega(T)$. We also note that $L_{\max}$ does not primarily constrain the fluctuations in demand. The fluctuation is only upper bounded during the period that determines $L_{\max}$, and there is no such constraint in the other rounds.

**Remark 3.** *It is straightforward to extend $L_{\max}$ to a high probability upper bound. In this case, we consider that there exists a parameter $0 < \delta < 1$ and $\mathbb{P}(\sum_{s=t}^{\min(t+L_{\max}-1, T+1)} d_s^i \geq D) \geq 1 - \delta/NT$ holds for any $i \in [N]$ and $t \in [T]$. This extension provides high-probability regret upper bounds. Furthermore, we note that $L_{\max}$ is essentially the same as the other parameters defined in Shi et al. (2016) and Hihat et al. (2023). In fact, the probabilistic extension is a generalization of them. We give a detailed discussion of this point in the appendix.* [4]

### 3.2 REGRET

We consider the following dynamic regret for OIO:

$$R_T(u_1, \ldots, u_T) = \sum_{t=1}^{T} \ell_t(y_t) - \sum_{t=1}^{T} \ell_t(u_t) \leq \sum_{t=1}^{T} \langle g_t, y_t - u_t \rangle. \tag{6}$$

Here $y_t \in \mathcal{C}(x_t)$, and $u_t \in \mathcal{C}(\mathbf{0})$. The major difficulty arises from the fact that $y_t$ and $u_t$ belong to the different feasible regions. Specifically, the feasible region of $u_t$ is always a superset of $y_t$'s feasible region, meaning that we employ a stronger comparator than that of the standard OCO problem. In the OIO setting, the feasible region of $y_t$ is affected by the previous decision; that is, the lower bound $x_t$ is constrained by $x_t^i \in [0, y_{t-1}^i]$ for all $i \in [N]$. Meanwhile, when we adopt a feasible comparator that satisfies $\max(0, u_t^i - d_t^i) \leq u_{t+1}^i$, the total path-length $P_T$ becomes bounded. We provide a detailed discussion in the appendix.

## 4 PROPOSED ALGORITHMS

---
**Algorithm 2** Online inventory optimization algorithm for dynamic environments
---
1: Set $L = 1$.
2: Initialize $x_1 = \mathbf{0}$ and $y_1 \in \mathcal{C}(x_1)$.
3: Initialize a base learner $\mathcal{E}(2L, T)$ with an initial state $\hat{y}_1 = y_1$.
4: **for** $t = 1, \ldots, T$ **do**
5:     Observe $g_t \in \partial \ell_t(y_t)$ and $x_{t+1}$ that satisfies $x_{t+1}^i \in [0, y_t^i]$ for all $i \in [N]$.
6:     Compute $\max \mathcal{L}_t$ where $\mathcal{L}_t$ is defined in Eq. (9).
7:     **if** $\max \mathcal{L}_t > L$ **then**
8:         Update $L \leftarrow 2L$ and restart $\mathcal{E}(2L, T)$ with the updated parameter $L$.
9:     **end if**
10:     Feed $g_t$ to $\mathcal{E}$ and receive a decision $\hat{y}_{t+1} \in \mathcal{C}(\mathbf{0})$.
11:     Update $y_{t+1} = \Pi_{\mathcal{C}(x_{t+1})}(\hat{y}_{t+1})$.
12: **end for**
---

Our algorithm employs a simple two-stage projection strategy, as described in Alg. 2. In each round $t$, the algorithm feeds $g_t$ into the base learner $\mathcal{E}$ and receives the decision $\hat{y}_{t+1} \in \mathcal{C}(\mathbf{0})$, which

---

[4]A simple example of $L_{\max}$ for a stochastic demand is the i.i.d. demand satisfying $d_t^i \in \{0, D/2\}$ with $\mathbb{P}(d_t^i \geq D/2) = 1/2$. In this case, Proposition 1 in the appendix implies that the high-probability upper bound of $L_{\max}$ is four plus an additional number of rounds that depends on the high-probability parameter.

only considers the warehouse capacity constraint (line 10). Then the algorithm projects it onto the feasible region with the carryover constraint: $\mathcal{C}(x_{t+1})$ (line 11). [5]

The organization of this section is as follows: We first discuss the properties of the projection $\Pi_{\mathcal{C}(x_{t+1})}$ in Section 4.1. Our key lemma is Lemma 1. By this lemma, we demonstrate that the regret upper bound of the decision $y_t$ can be reduced to that of the base learner's decision $\hat{y}_t$. Furthermore, we show that the carryover stock constraint leads to a switching cost for $\hat{y}_t$ in the base learner's regret. In Section 4.2, we provide a regret guarantee for a general base learner in Theorems 2. Finally, in Section 4.3, we introduce SOCO algorithms with a dynamic regret guarantee and present its regret upper bound in Theorem 4. [6]

## 4.1 PROJECTION PROPERTY

Our analysis is based on time-periods called *cycles*. For each item $i$, a cycle is defined by the period during which $\hat{y}_t^i$ cannot be realized due to the carryover stock $x_t^i$, resulting in $y_t^i > \hat{y}_t^i$. This is formally expressed as follows:

**Definition 2** (Cycle). Let $\mathcal{S}_i \subset [T]$ denote the set of rounds $t \in [T]$ such that $y_t^i \leq \hat{y}_t^i$. Suppose the elements $t \in \mathcal{S}_i$ are indexed in strictly increasing order as $t_1 < t_2 < \cdots < t_{|\mathcal{S}_i|}$. We refer to the period $t_k, t_k + 1, \ldots, t_{k+1} - 1$ for $t_k \in \mathcal{S}_i$ as *the $k$-th cycle of item $i$*, and define the length of the $k$-th cycle as $L_k^i := t_{k+1} - t_k$, where we set $t_{|\mathcal{S}_i|+1} = T + 1$.

Then, the following key lemma holds in our OIO setting:

**Lemma 1.** *For any base learner $\mathcal{E}$, Alg. 2 ensures*

$$\sum_{t=1}^{T} \langle g_t, y_t - \hat{y}_t \rangle \leq 2G \sum_{t=1}^{T} \left( \max_{i \in [N]} L_t^i \right) \|\hat{y}_t - \hat{y}_{t+1}\|_1 , \tag{7}$$

*where $L_t^i$ is the current cycle length for item $i$, that is, $L_k^i$ that satisfies $t_k \leq t < t_{k+1}$ for $t_k, t_{k+1} \in \mathcal{S}_i$.*

**Remark 4.** *Lemma 1 shows that, under our two-stage projection strategy, OIO is linked to SOCO (Lin et al., 2011; Zhang et al., 2021; 2022c;a), eliminating the difficulty for the dynamic carryover stock constraint in the OIO setting.*

In fact, under Alg. 2, the regret is bounded as

$$R_T \leq \sum_{t=1}^{T} \left( \langle g_t, \hat{y}_t - u_t \rangle + 2GL_t^*\|\hat{y}_t - \hat{y}_{t+1}\|_1 \right) , \tag{8}$$

where $L_t^* = \max_{i \in [N]} L_t^i$. The right-hand side is interpreted as the dynamic regret for SOCO problem for the base learner $\mathcal{E}$, where for every $t \in [T]$, $\mathcal{E}$ chooses $\hat{y}_t \in \mathcal{C}(\mathbf{0})$ and suffers loss $\langle g_t, \hat{y}_t \rangle$ with switching cost of $2GL_{t-1}^*\|\hat{y}_{t-1} - \hat{y}_t\|_1$. The main difference from the standard SOCO is the coefficient $L_t^*$, which is time-dependent and delayed in observability; it becomes observable only after the cycle for each item at time $t$ is completed. [7] We propose an improved algorithm that works without prior knowledge of the switching cost in the next section.

## 4.2 DOUBLING TRICK FOR THE UNKNOWN SWITCHING COST

We address the unknown switching cost in Eq. (8) by introducing a doubling trick for $L_t^*$. In Alg. 2, as described in lines 7 to 9, our algorithm restarts the base learner $\mathcal{E}$ with a new parameter $L$ by

---

[5] We initialize $x_1$ as $\mathbf{0}$ and the beginning of the first cycle is $t = 1$. We note that our algorithm can be applied for the $x_1 \neq \mathbf{0}$ case, incurring an additional regret of at most $GDL_{\max}$ by adopting the zero-order strategy until the inventory level reaches $\mathbf{0}$.

[6] All omitted proofs are given in the appendix. We also omit the high-probability regrets for the sake of clarity, since the extension is rather straightforward. See Remark 5 in the appendix for details.

[7] Another difference is that the switching cost appears as $\ell_1$-norm instead of the $\ell_2$-norm. We track this impact in the regret analyses.

comparing the current parameter and the maximum observed cycle length $\max \mathcal{L}_t$. Here, we define the set of the observed cycle lengths at round $t$ as

$$\mathcal{L}_t := \bigcup_{i \in [N]} \left\{ L_1^i, \ldots, L_{k-1}^i, t - t_k + 1 \mid t_k \leq t < t_{k+1}, t_k, t_{k+1} \in \mathcal{S}_i \right\}, \tag{9}$$

where $t - t_k + 1$ indicates the lower bound of the current cycle length. We note that in Alg. 2, we do not store $\mathcal{L}_t$ explicitly because the algorithm only needs $\max \mathcal{L}_t$, which can be tracked with $\mathcal{O}(N)$ memory. For the regret upper bound analysis, we use the following property of the cycle length:

**Lemma 2.** *The cycle length is upper bounded by the sell-out period $L_{\max}$.*

We assume that the base learner is an algorithm $\mathcal{E}(L, T)$ with an input parameter $L$ and $T$ that provides a regret upper bound of

$$\sum_{t=1}^{T} \left( \langle g_t, \hat{y}_t - u_t \rangle + GL \|\hat{y}_t - \hat{y}_{t+1}\|_1 \right) \leq \mathcal{R}_{L,T}^{\mathcal{E}(L,T)} \tag{10}$$

for any series of $\{g_t\}_{t=1}^{T}$.

Then, the following regret upper bound holds for Alg. 2.

**Theorem 2.** *Assume that under algorithm $\mathcal{E}(L, T)$, the regret upper bound $\mathcal{R}_{L,T}^{\mathcal{E}(L,T)}$ can be decomposed into $\mathcal{R}_{L,T}^{\mathcal{E}(L,T)} = L^\alpha \mathcal{R}(T)$ for $\alpha > 0$ and the switching cost is bounded by $\|\hat{y}_t - \hat{y}_{t+1}\|_1 \leq \mathcal{O}(L^{-\beta})$ for $\beta \geq 0$. Then, Alg. 2 ensures*

$$R_T \leq C(\alpha) \mathcal{R}_{2L_{\max},T}^{\mathcal{E}(2L_{\max},T)} + \Delta(L_{\max}, \beta),$$

*where $C(\alpha)$ is an $\alpha$-dependent factor and $\Delta(L_{\max}, \beta)$ denotes $\mathcal{O}(L_{\max}^{2-\beta})$ if $\beta < 1$, $\mathcal{O}(L_{\max} \log L_{\max})$ if $\beta = 1$, and $\mathcal{O}(L_{\max})$ if $\beta > 1$.*

### 4.3 ALGORITHMS FOR THE BASE LEARNER

In this section we introduce algorithms for SOCO that can be used as the base learner in Alg. 2. First, we introduce the standard Online Gradient Descent algorithm (OGD) described in Alg. 3.

**Theorem 3.** *Assume $T \geq L_{\max}(3 + P_T/D)$. In Alg. 2, the base learner Alg. 3 with an $L$-parameterized learning rate $\eta = \sqrt{\frac{2D(3D+P_T)}{G^2(\sqrt{N}L+1/2)T}}$ ensures $R_T \leq \mathcal{O}(\sqrt{L_{\max}(1 + P_T)T} + L_{\max} \log L_{\max})$.*

To obtain the optimal regret order, we must know $P_T$ *a priori* when setting the learning rate $\eta$. This parameter depends on the characteristics of the future demands and is sometimes difficult to determine in advance.

Recently, Zhang et al. (2022a) have proposed the Smoothed Online Gradient Descent algorithm (SOGD). In the algorithm, the meta-algorithm sequentially aggregates multiple experts' decisions, where $k$-th decision in the sequence is obtained by combining $k$-th expert's decision $\hat{y}_{t+1}^k$ and $k-1$-th combined decision $\hat{v}_{t+1}^{k-1}$ via the $k$-th combiner $\mathcal{B}^k$. The combiner combines the two inputs with a weight $p_{t+1}$ that is adaptively computed by Discounted-Normal-Predictor (Kapralov & Panigrahy, 2011) with conservative updating using bit sequences of

$$b_t^k := \frac{\langle g_t, \hat{v}_t^{k-1} - \hat{y}_t^k \rangle + GL(\|\hat{v}_t^{k-1} - \hat{v}_{t+1}^{k-1}\|_1 - \|\hat{y}_t^k - \hat{y}_{t+1}^k\|_1)}{6GDN^{1/4}\sqrt{L}} \tag{11}$$

as described in line 5 to 11 in Alg. 4. The meta-algorithm uses $K$-th decision as the output.

**Theorem 4.** *Assume $T \geq \sqrt{L_{\max}}(\log_2 T + e)$. In Alg. 2, the base learner Alg. 5 ensures*

$$R_T \leq \mathcal{O}(\sqrt{L_{\max}(1 + P_T)T \log T} + L_{\max} \log L_{\max}).$$

**Algorithm 3** Online gradient descent

**Require:** Learning rate $\eta$.
1: **for** $t = 1, \ldots, T$ **do**
2:     Receive a subgradient $g_t$.
3:     Return $\hat{y}_{t+1} = \Pi_{\mathcal{C}(\mathbf{0})}(\hat{y}_t - \eta g_t)$.
4: **end for**

**Algorithm 4** $k$-th combiner

**Require:** Two parameters: $n^k$ and $L$.
1: Initialize $z_1 = 0$.
2: Set $\tilde{g}(z) := \sqrt{\frac{n^k}{8}} \frac{1}{T} \text{erf}(\frac{z}{\sqrt{8n^k}}) e^{z^2/16n^k}$.
3: Compute $U(n^k) := \tilde{g}^{-1}(1)$
4: **for** $t = 1, \ldots, T$ **do**
5:     Receive $\hat{v}_{t+1}^{k-1}, \hat{y}_{t+1}^k$, and $g_t$.
6:     Compute $b_t^k$ by Eq. (11).
7:     **if** $z_t \in [0, U(n^k)]$ or $(z_t < 0) \cap (b_t^k > 0)$ or $(z_t > U(n^k)) \cap (b_t^k < 0)$ **then**
8:         $z_{t+1} = (1 - 1/n^k)z_t + b_t^k$.
9:     **else**
10:        $z_{t+1} = (1 - 1/n^k)z_t$.
11:    **end if**
12:    $p_{t+1}^k = \Pi_{[0,1]}(\tilde{g}(z_{t+1}))$
13:    Return $\hat{v}_{t+1}^k = (1 - p_{t+1}^k)\hat{v}_{t+1}^{k-1} + p_{t+1}^k \hat{y}_{t+1}^k$.
14: **end for**

**Algorithm 5** Smoothed online gradient descent (Zhang et al., 2022a)

**Require:** $L > 0$.
1: Set $K = \lfloor \log_2 \frac{T}{32 \max(L,1) \log T} \rfloor + 1$.
2: **for** $k = 1, \ldots, K$ **do**
3:     Set $n^k = T2^{1-k}$
4:     Initialize $k$-th instance $\mathcal{A}^k$, which is Alg. 3 with the learning rate of $\eta^k = 2D/G\sqrt{1/(2\sqrt{N}L+1)n^k}$.
5:     Initialize $k$-th combiner $\mathcal{B}^k$, which is Alg. 4 with the input parameters of $n^k$ and $L$.
6: **end for**
7: **for** $t = 1, \ldots, T$ **do**
8:     Receive a subgradient $g_t$.
9:     **for** $k = 1, \ldots, K$ **do**
10:        **if** k = 1 **then**
11:            $\hat{v}_{t+1}^1 \leftarrow \mathcal{A}^1(g_t)$.
12:        **else**
13:            $\hat{y}_{t+1}^k \leftarrow \mathcal{A}^k(g_t)$.
14:            $\hat{v}_{t+1}^k \leftarrow \mathcal{B}^k(\hat{v}_{t+1}^{k-1}, \hat{y}_{t+1}^k, g_t)$.
15:        **end if**
16:    **end for**
17:    Return $\hat{y}_{t+1} = \hat{v}_{t+1}^K$.
18: **end for**

We note that $\mathcal{O}(L_{\max} \log L_{\max})$ in Theorems 3 and 4 is the overhead incurred by the doubling-trick mechanism and is subdominant for a broad range of horizons, e.g., $T > L_{\max} \log^2 L_{\max}$.

We finally comment on the computational overhead of our algorithm. In Alg. 5, in order to suppress the dynamic regret upper bound, we adopt a meta-algorithm that incurs a computational cost of $\mathcal{O}(KT) = \mathcal{O}(T \log T)$, which is more expensive than OGD by a factor of $\log T$. Such overhead is common in the OCO or SOCO setting, when using meta-algorithms for non-stationary environments. On the other hand, the doubling trick in Alg. 2 restarts the base learner at most $\mathcal{O}(\log L_{\max})$ times, resulting in an additional $\mathcal{O}(\log L_{\max})$ initialization overhead which does not affect the per-round computational cost.

## 5 LOWER BOUND

In this section, we discuss the optimality of our regret analysis. In OCO, Zhang et al. (2018b) have established the $\Omega(\sqrt{(1 + P_T)T})$ lower bound. Our regret upper bound matches this lower bound up to a logarithmic factor. On the other hand, we also have a $\sqrt{L_{\max}}$ factor in our bound. The following theorem ensures this optimality.

**Theorem 5.** *For any algorithm $\mathcal{A}$, there exists some sequence $\{g_t\}_t$ and some $u \in \mathcal{C}(\mathbf{0})$ such that*

$$\sum_{t=1}^{T} \langle g_t, y_t - u \rangle = \Omega(GD\sqrt{L_{\max}T}),$$

*where $\{y_t\}_{t=1}^T$ is the sequence of the outputs by $\mathcal{A}$.*

As a byproduct, this lower bound provides the optimality of the $\sqrt{L}$ factor in the OGD and SOGD algorithms for the SOCO setting. This is because if there were an algorithm that can be improved upon, it can break the lower bound of OIO by adopting it as the base learner of our algorithm.

**Corollary 1.** *For SOCO with regret of $\tilde{R}_T(L)$, its lower bound is $\Omega(\sqrt{LT})$.*

In our study, OIO and SOCO are found to be connected, which provides an intriguing example of how one lower bound can constrain the other.

## 6   CONCLUSIONS AND LIMITATIONS

In this paper, we propose an algorithm for OIO with a near-optimal dynamic regret guarantee. We connect OIO to SOCO through a simple two-stage projection and the dynamic regret bound combining an algorithm for SOCO and doubling trick for unknown $L_{\max}$.

There are several interesting prospects for future investigation. First, the problem setting does not take into account the lead time and fixed-order costs. For i.i.d. demand, there are studies addressing these settings (Zhang et al., 2020; Agrawal & Jia, 2022; Yuan et al., 2021). The extension to dynamic environments is an interesting direction for future research. Secondly, we assume a linear capacity constraint as described in Eq. (3). This assumption is critical to the proof of Lemmas 5 and 6. Although we believe that it is possible to extend this assumption to a more general convex set, we leave it for future work.

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

## A   DISCUSSION ON $L_{\max}$

Existing works introduce the least amount of demands in each round. Shi et al. (2016) assumes minimum demand. Hihat et al. (2023) introduces parameters $\rho$ and $\mu$ and assumes $\mathbb{P}\left[d_t^i \geq \rho\right] \geq \mu$ holds for all $t \in [T]$ almost surely. We here see the relation between Assumption 10 in Hihat et al. (2023) and Remark 3 in our paper.

**Proposition 1.** *If Assumption 10 in Hihat et al. (2023) holds, then our assumption in Remark 3 holds. That is, if one has $\mu$ and $\rho$ such that $\mathbb{P}\left[d_t^i \geq \rho\right] \geq \mu$ holds for all $t \in [T]$ almost surely, then there exists $L_{\max}$ such that $\mathbb{P}\left(\sum_{s=t}^{\min(t+L_{\max}-1, T+1)} d_s^i \geq D\right) \geq 1 - \delta/NT$ holds for any $i \in [N]$ and $t \in [T]$.*

*Proof.* We denote $L_{\max}$ by $L$. Suppose Assumption 10 in Hihat et al. (2023) holds. Then, we have

$$\mathbb{P}\left[d_t^i \geq \rho\right] \geq \mu$$

for all $t \in [T]$ almost surely. By Markov's inequality, we obtain

$$\mathbb{P}\left[d_t^i \geq \rho\right] \leq \frac{\mathbb{E}\left[d_t^i\right]}{\rho},$$

and thus $\mathbb{E}\left[d_t^i\right] \geq \rho\mu$ holds. Our aim is to obtain the number of rounds necessary for making the inventory sold out with the probability at least $1 - \delta/NT$ for each cycle (by the fact that there exist at most $T$ cycles from $t = 1$ to $t = T$ for all items and technique of the union bound). Since we assume $x_{T+1}^i = 0$ for all $i \in [N]$, we consider $L$ consecutive rounds only. Hereafter, we consider some fixed $i \in [N]$. Let us denote

$$X_t = \begin{cases} 1 & (d_t^i \geq \rho) \\ 0 & (d_t^i < \rho) \end{cases}$$

and

$$Y_t = \sum_{s=1}^{t} (X_s - \mathbb{E}[X_s]).$$

By applying Azuma–Hoeffding inequality, we obtain

$$\mathbb{P}\left(\sum_{t=1}^{L} X_t \leq L\mu - \varepsilon\right) \leq \mathbb{P}(Y_L - Y_0 \leq -\varepsilon)$$

$$= \mathbb{P}\left(\sum_{t=1}^{L} X_t \leq \mathbb{E}\left[\sum_{t=1}^{L} X_t\right] - \varepsilon\right)$$

$$\leq \exp\left(-\frac{\varepsilon^2}{2L}\right).$$

From

$$\exp\left(-\frac{\varepsilon^2}{2L}\right) \leq \frac{\delta}{NT},$$

we obtain

$$\varepsilon \geq \sqrt{2L \log \frac{NT}{\delta}}.$$

Therefore, $\sum_{t=1}^{L} X_t \leq L\mu - \sqrt{2L \log \frac{NT}{\delta}}$ holds with probability at least $1 - \delta/NT$. If demand larger than or equal to $\rho$ occurs at least $D/\rho$ times, then the inventory becomes sold out. Thus, the condition for $L$ is

$$\frac{D}{\rho} \leq L\mu - \sqrt{2L \log \frac{NT}{\delta}}.$$

Let us denote $w = \sqrt{L\mu}$, $a = \sqrt{\frac{2 \log \frac{NT}{\delta}}{\mu}}$, and $b = D/\rho$, then we obtain

$$\frac{D}{\rho} \leq L\mu - \sqrt{2L \log \frac{NT}{\delta}} \iff w^2 - aw \geq b$$

$$\iff \left( w - \frac{a}{2} \right)^2 \geq \frac{a^2}{4} + b$$

$$\Longleftarrow w - \frac{a}{2} \geq \sqrt{\frac{a^2}{4} + b}$$

$$\iff w \geq \frac{a}{2} + \sqrt{\frac{a^2}{4} + b}.$$

Then,

$$w \geq \frac{a}{2} + \sqrt{\frac{a^2}{4} + b} \iff \sqrt{L\mu} \geq \sqrt{\frac{\log \frac{NT}{\delta}}{2\mu}} + \sqrt{\frac{\log \frac{NT}{\delta}}{\mu} + \frac{D}{\rho}}$$

$$\iff L\mu \geq \left( \sqrt{\frac{\log \frac{NT}{\delta}}{2\mu}} + \sqrt{\frac{\log \frac{NT}{\delta}}{\mu} + \frac{D}{\rho}} \right)^2$$

$$\Longleftarrow L\mu \geq \frac{\log \frac{NT}{\delta}}{\mu} + \frac{2 \log \frac{NT}{\delta}}{\mu} + \frac{2D}{\rho}$$

holds, where the last part utilizes $(\alpha + \beta)^2 \leq 2(\alpha^2 + \beta^2)$, $\forall \alpha, \beta \in \mathbb{R}$. Therefore, if one adopts $L$ satisfying

$$L\mu \geq \frac{3 \log \frac{NT}{\delta}}{\mu} + \frac{2D}{\rho} \iff L \geq \frac{2D}{\rho\mu} + \frac{3 \log \frac{NT}{\delta}}{\mu^2},$$

the inventory becomes sold out in at most $L$ rounds with the probability at least $1 - \delta/NT$. $\qquad \square$

## B    EXTENSION TO THE HIGH-PROBABILITY REGRET

**Remark 5.** *The probabilistic definition for $L_{\max}$ in Remark 3 ensures that $L_k^i$ satisfies $L_k^i \leq L_{\max}$ with probability of $1 - \delta/NT$. Given this definition, our regret upper bounds hold when all $L_k^i$ satisfy $L_k^i \leq L_{\max}$ in $t \in [T]$. Applying the union bound over all cycles and items, we bound its probability at least $1 - \delta$. Therefore, using the probabilistic expression for $L_{\max}$, our results naturally extend to high-probability regrets, maintaining the same order of bounds with a probability of $1 - \delta$.*

## C    ORDER ESTIMATION OF $P_T$

**Proposition 2.** *Under the feasible comparator that satisfies $\max(0, u_t^i - d_t^i) \leq u_{t+1}^i$, $P_T$ is upper bounded by $ND + 2 \sum_{i=1}^{N} \sum_{t=1}^{T} d_t^i$.*

*Proof.* For clarity, we first consider the single-item scenario. Consider a set $\mathcal{A} = \{t \in \{2, \ldots, T\} \mid u_{t-1} \geq u_t\}$, and write $P_T$ as

$$P_T = \sum_{t=2}^{T} |u_{t-1} - u_t| = \sum_{t \in \mathcal{A}} (u_{t-1} - u_t) + \sum_{t \in \{2, \ldots, T\} \setminus \mathcal{A}} (u_t - u_{t-1}). \tag{12}$$

The first term is upper bounded by the demand series $\{d_t\}$ as

$$u_{t-1} - u_t \leq d_{t-1}, \tag{13}$$

because the feasible space of $u_t$ is constrained by the carryover stock as $u_t \geq \max(u_{t-1} - d_{t-1}, 0)$. On the other hand, the second term can be bounded by the first term as follows:

$$-D \leq u_1 - u_T = \sum_{t=2}^{T}(u_{t-1} - u_t) = \sum_{t \in \mathcal{A}}(u_{t-1} - u_t) - \sum_{t \in \{2,...,T\} \setminus \mathcal{A}}(u_t - u_{t-1}). \tag{14}$$

Combining these inequalities, we have

$$P_T \leq 2\sum_{t \in \mathcal{A}}(u_{t-1} - u_t) + D \leq 2\sum_{t \in \mathcal{A}} d_{t-1} + D \leq 2\sum_{t=1}^{T} d_t + D. \tag{15}$$

The bound in the multi-item case can be obtained straightforwardly as the sum of the bounds for each item, which concludes the proof. $\square$

We also note that the ideal feasible comparator typically yields $P_T = \sum_{t=2}^{T}\|d_{t-1} - d_t\|_1$. This is because, in most inventory system without lead-time, the ideal order-up-to decision $\{u_t\}$ matches the demand $\{d_t\}$, which incurs neither lost-sales loss nor holding costs.

# D PROOFS OF THE LEMMAS IN SECTIONS 4.1 AND 4.2

## D.1 LEMMAS ON THE PROJECTION OPERATOR $\Pi_{\mathcal{C}(x)}$

In this section, we provide lemmas regarding the relationships that hold between $\hat{y} \in \mathcal{C}(\mathbf{0})$ and $y = \Pi_{\mathcal{C}(x)}(\hat{y})$ where $x \in \mathcal{C}(\mathbf{0})$. We note that $y$, $\hat{y}$, and $x$ do not necessarily depend on $t$; in other words, we do not assume that they are elements of a $t$-dependent series resulting from a particular algorithm or environment.

For the subsequent proofs, we define the set of the item index $\mathcal{I}$ and its complement as $\mathcal{I} := \{i \in [N] \mid y^i \leq \hat{y}^i\}$, and $\overline{\mathcal{I}} := [N] \setminus \mathcal{I} = \{i \in [N] \mid y^i > \hat{y}^i\}$, respectively. Recall that $\mathcal{C}(x) \subset \mathcal{C}(\mathbf{0})$ and the projection $y = \Pi_{\mathcal{C}(x)}(\hat{y})$ is equal to $y = \arg\min_{y' \in \mathcal{C}(x)} \|y' - \hat{y}\|_2^2$.

**Lemma 3.** *For $i \in \overline{\mathcal{I}}$, $y^i = x^i > 0$.*

*Proof.* We divide the proof into three cases regarding $\hat{y}$; (i) For $\hat{y} \in \mathcal{C}(x)$, it is obvious that $y = \hat{y}$ holds. (ii) For $\hat{y} \notin \mathcal{C}(x)$ and $\hat{y}^i < x^i$, we observe $y^i = x^i > \hat{y}^i$. This is because if we have some $\epsilon > 0$ and $y^i = x^i + \epsilon$, decreasing $\epsilon$ to zero decreases the objective function without violating the constraint, which contradicts the minimality of $y$. We also note that $x^i > 0$ in this case because $\hat{y}^i \geq 0$. (iii) Finally, for $\hat{y} \notin \mathcal{C}(x)$ and $\hat{y}^i \geq x^i$, we observe $y^i \leq \hat{y}^i$. This is because if we have some $\epsilon > 0$ and $y^i = \hat{y}^i + \epsilon$, decreasing $\epsilon$ to zero decreases the objective function without violating the constraint, which contradicts the minimality of $y$. In summary, $y^i > \hat{y}^i$ only occurs in the case of (ii), which leads to $y^i = x^i > 0$. $\square$

**Lemma 4.** *If there exists an $i^* \in [N]$ that satisfies $y^{i^*} < \hat{y}^{i^*}$, then $\sum_{i \in \mathcal{I}} y^i = D - \sum_{j \in \overline{\mathcal{I}}} x^j$.*

*Proof.* From Lemma 3, it is obvious $y^j = x^j$ for $j \in \overline{\mathcal{I}}$. Therefore, $y = \Pi_{\mathcal{C}(x)}(\hat{y})$ implies that $y$ minimizes $\sum_{i \in \mathcal{I}}(y^i - \hat{y}^i)^2$ satisfying $y^i \leq \hat{y}^i$ and $\sum_{i \in \mathcal{I}} y^i \leq D - \sum_{j \in \overline{\mathcal{I}}} x^j$. Assume that $\sum_{i \in \mathcal{I}} y^i < D - \sum_{j \in \overline{\mathcal{I}}} x^j$. Then, we can increase $y^{i^*}$ to $\hat{y}^{i^*}$ without violating the constraint, which decreases the objective function and contradicts the minimality of $y$. $\square$

## D.2 PROOF OF LEMMA 1

To prove Lemma 1, we use the following two lemmas for the cycle property. Let $\mathcal{I}_t$ be the set of items such that $t$ is the initial part of the cycle, i.e., $\mathcal{I}_t := \{i \in [N] \mid y_t^i \leq \hat{y}_t^i\}$. Note that $\overline{\mathcal{I}}_t := [N] \setminus \mathcal{I}_t = \{i \in [N] \mid y_t^i > \hat{y}_t^i\}$ is the set of items in the later part of the cycle. Then, the following lemmas hold.

**Lemma 5.** *For any $t \in [T]$, $\sum_{i \in \mathcal{I}_t} \hat{y}_t^i - y_t^i \leq \sum_{i \in \overline{\mathcal{I}}_t} y_t^i - \hat{y}_t^i$.*

**Lemma 6.** *For any $k \in [K^i]$ where $K^i$ is the number of cycles for item $i$: $K^i := |\mathcal{S}_i|$ and $s \in [L_k^i - 1]$, $y_{t_k+s}^i - \hat{y}_{t_k+s}^i \leq \sum_{s'=0}^{s-1} \hat{y}_{t_k+s'}^i - \hat{y}_{t_k+s'+1}^i$.*

*Proof of Lemma 1.* We divide the left-hand side of Eq. (7) into the initial and later parts of the cycle:

$$\sum_{t=1}^{T} \langle g_t, y_t - \hat{y}_t \rangle = \sum_{t=1}^{T} \sum_{i \in \mathcal{I}_t} g_t^i(y_t^i - \hat{y}_t^i) + \sum_{i \in \overline{\mathcal{I}}_t} g_t^i(y_t^i - \hat{y}_t^i). \tag{16}$$

For the first term, from Lemma 5, the following inequality holds:

$$\sum_{t=1}^{T} \sum_{i \in \mathcal{I}_t} g_t^i(y_t^i - \hat{y}_t^i) \leq \sum_{t=1}^{T} \|g_t\|_\infty \sum_{i \in \mathcal{I}_t} (\hat{y}_t^i - y_t^i) \overset{\text{Lemma 5}}{\leq} \sum_{t=1}^{T} \|g_t\|_\infty \sum_{i \in \overline{\mathcal{I}}_t} (y_t^i - \hat{y}_t^i), \tag{17}$$

where we use $y_t^i \leq \hat{y}_t^i$ for $i \in \mathcal{I}_t$ in the first inequality. This inequality suggests the following statement: *the contributions from the initial part of the cycles in all items are bounded by the contributions from the later parts of the cycles in all items.* Therefore, the proof is completed by evaluating the contributions from the later parts of the cycles, i.e., the second term in Eq. (16).

For the second term in Eq. (16), using Lemma 6, we have

$$\sum_{t=1}^{T} \sum_{i \in \overline{\mathcal{I}}_t} g_t^i(y_t^i - \hat{y}_t^i) = \sum_{i=1}^{N} \sum_{k=1}^{K^i} \sum_{s=1}^{L_k^i - 1} g_{t_k^i+s}^i(y_{t_k^i+s}^i - \hat{y}_{t_k^i+s}^i)$$

$$\leq \sum_{i=1}^{N} \sum_{k=1}^{K^i} \sum_{s=1}^{L_k^i - 1} \|g_{t_k^i+s}\|_\infty (y_{t_k^i+s}^i - \hat{y}_{t_k^i+s}^i)$$

$$\overset{\text{Lemma 6}}{\leq} \sum_{i=1}^{N} \sum_{k=1}^{K^i} \sum_{s=1}^{L_k^i - 1} \|g_{t_k^i+s}\|_\infty \sum_{s'=0}^{s-1} (\hat{y}_{t_k^i+s'}^i - \hat{y}_{t_k^i+s'+1}^i), \tag{18}$$

where we refer to the definition of the summation of the later parts of the cycle for the first equality. Combining Eq. (16), Eq. (17), and Eq. (18), we finally have

$$\sum_{t=1}^{T} \langle g_t, y_t - \hat{y}_t \rangle \overset{\text{Eq. (17)}}{\leq} 2 \sum_{i=1}^{N} \sum_{k=1}^{K^i} \sum_{s=1}^{L_k^i - 1} \|g_{t_k^i+s}\|_\infty (y_{t_k^i+s}^i - \hat{y}_{t_k^i+s}^i)$$

$$\overset{\text{Eq. (18)}}{\leq} 2 \sum_{i=1}^{N} \sum_{k=1}^{K^i} \sum_{s=1}^{L_k^i - 1} \|g_{t_k^i+s}\|_\infty \sum_{s'=0}^{s-1} (\hat{y}_{t_k^i+s'}^i - \hat{y}_{t_k^i+s'+1}^i)$$

$$\leq 2G \sum_{i=1}^{N} \sum_{k=1}^{K^i} \sum_{s=1}^{L_k^i - 1} \sum_{s'=0}^{s-1} (\hat{y}_{t_k^i+s'}^i - \hat{y}_{t_k^i+s'+1}^i)$$

$$\overset{\text{Lemma 7}}{=} 2G \sum_{i=1}^{N} \sum_{t=1}^{T} \left( L_{\kappa^i(t)}^i - (t - t_{\kappa^i(t)}) - 1 \right) (\hat{y}_t^i - \hat{y}_{t+1}^i)$$

$$\leq 2G \sum_{t=1}^{T} \left( \max_{i \in [N]} L_{\kappa^i(t)}^i \right) \|\hat{y}_t - \hat{y}_{t+1}\|_1$$

$$\leq 2G \sum_{t=1}^{T} L_t^* \|\hat{y}_t - \hat{y}_{t+1}\|_1.$$

In the fourth line, we apply Lemma 7. This concludes the proof. $\square$

### D.3 PROOF OF LEMMA 5

*Proof.* First, we consider the case $\mathcal{I} = [N]$. In this case, we observe $\hat{y}^i - y^i = 0$ for all $i \in [N]$. This can be proved as follows: If we have non-empty set $\mathcal{I}' := \{i \in [N] \mid y^i < \hat{y}^i\}$, we can write $y^j = \hat{y}^j - \epsilon^j$ where $\epsilon^j > 0$ for $j \in \mathcal{I}'$. Then, $\sum_{i \in [N]} y^i = \sum_{i \in [N]} \hat{y}^i - \sum_{j \in \mathcal{I}'} \epsilon^j \leq D - \sum_{j \in \mathcal{I}'} \epsilon^j$. Therefore, decreasing $\epsilon^j$s to zero decreases the objective function without violating the constraint, which contradicts the minimality of $y$.

Then, we consider the case $\mathcal{I} \neq [N]$. If all $i \in \mathcal{I}$ satisfies $y^i = \hat{y}^i$, then $\sum_{i \in \mathcal{I}} (\hat{y}^i - y^i) = 0$ and the inequality holds. Otherwise, from Lemma 4, we have $\sum_{i \in \mathcal{I}} y^i = D - \sum_{j \in \overline{\mathcal{I}}} x^j$ and

$$
\begin{aligned}
\sum_{i \in \mathcal{I}} \hat{y}^i - y^i &= \sum_{i \in \mathcal{I}} \hat{y}^i - D + \sum_{j \in \overline{\mathcal{I}}} x^j \\
&= \sum_{i \in [N]} \hat{y}^i - D + \sum_{j \in \overline{\mathcal{I}}} (x^j - \hat{y}^j) \\
&\leq \sum_{j \in \overline{\mathcal{I}}} (x^j - \hat{y}^j) \\
&\overset{\text{Lemma 3}}{=} \sum_{j \in \overline{\mathcal{I}}} (y^j - \hat{y}^j) \,.
\end{aligned}
$$

In the last inequality, we use $\sum_{i \in [N]} \hat{y}^i \leq D$ because $\hat{y} \in \mathcal{C}(\mathbf{0})$. $\qquad\square$

### D.4 PROOF OF LEMMA 6

*Proof.* For the sake of brevity, we omit index $i$ of $t_k^i$, $L_k^i$, and $K^i$ when it is clear from the context. Consider the summation in the $k$-th cycle for item $i$:

$$
g_{t_k}^i(y_{t_k}^i - \hat{y}_{t_k}^i) + g_{t_k+1}^i(y_{t_k+1}^i - \hat{y}_{t_k+1}^i) + \cdots + g_{t_k+L_k-1}^i(y_{t_k+L_k-1}^i - \hat{y}_{t_k+L_k-1}^i) \,.
$$

From the definition of the $k$-th cycle, we have

$$
y_{t_k}^i \leq \hat{y}_{t_k}^i \,. \tag{19}
$$

Moreover, for $s = 1, \ldots, L_k - 1$, because $y_{t_k+s}^i > \hat{y}_{t_k+s}^i$, we have $y_{t_k+s}^i = x_{t_k+s}^i > 0$ from Lemma 3. Thus, the following order property holds:

$$
y_{t_k+s-1}^i \overset{\text{Eq. (1)}}{\geq} x_{t_k+s}^i \overset{\text{Lemma 3}}{=} y_{t_k+s}^i > \hat{y}_{t_k+s}^i \geq 0 \,, \tag{20}
$$

for $s = 1, \ldots, L_k - 1$. Using the above properties, for cycles of $L_k \geq 2$, the following upper bound holds:

$$
\begin{aligned}
y_{t_k+s}^i - \hat{y}_{t_k+s}^i = x_{t_k+s}^i - \hat{y}_{t_k+s}^i &\leq y_{t_k+s-1}^i - \hat{y}_{t_k+s}^i \\
&= (y_{t_k+s-1}^i - \hat{y}_{t_k+s-1}^i) + (\hat{y}_{t_k+s-1}^i - \hat{y}_{t_k+s}^i) \\
&= \ldots \\
&= (y_{t_k}^i - \hat{y}_{t_k}^i) + \sum_{s'=0}^{s-1} (\hat{y}_{t_k+s'}^i - \hat{y}_{t_k+s'+1}^i) \\
&\overset{\text{Eq. (19)}}{\leq} \sum_{s'=0}^{s-1} (\hat{y}_{t_k+s'}^i - \hat{y}_{t_k+s'+1}^i) \,,
\end{aligned}
$$

which concludes the proof. $\qquad\square$

### D.5 THE OTHER TECHNICAL LEMMA FOR LEMMA 1

**Lemma 7.** *Suppose round $1, \ldots, T$ is divided into $K$ segments of lengths $L_1, \ldots, L_K$ that satisfy $1 \leq L_k \leq T \, \forall k \in [K]$ and $\sum_{k=1}^K L_k = T$. Let us define a function $\kappa : [T] \to [K]$ which maps each*

*round $t \in [T]$ to the segment $k \in [K]$ that $t$ belongs to, i.e., $\kappa(t) := \min_{k \in [K]} k$ s.t., $\sum_{k'=1}^{k} L_{k'} \geq t$. Then, for any series $a_1, \ldots, a_T$ and $b_1, \ldots, b_K$, the following equality holds:*

$$\sum_{k=1}^{K} \sum_{s=1}^{L_k-1} \sum_{s'=0}^{s-1} a_{t_k+s'} b_k = \sum_{t=1}^{T} a_t b_{\kappa(t)} [L_{\kappa(t)} - (t - t_{\kappa(t)}) - 1]_+ ,$$

*where $t_k := \sum_{k'=1}^{k-1} L_{k'} + 1$ is the initial round of $k$-th segment and $[x]_+ := x \mathbb{I}[x \geq 0]$.*

*Proof.*

$$
\begin{aligned}
\sum_{k=1}^{K} \sum_{s=1}^{L_k-1} \sum_{s'=0}^{s-1} a_{t_k+s'} b_k &= \sum_{t=1}^{T} \sum_{k=1}^{K} \sum_{s=1}^{L_k-1} \sum_{s'=0}^{s-1} a_t b_k \mathbb{I}[t = t_k + s'] \\
&= \sum_{t=1}^{T} \sum_{k=1}^{K} \sum_{s=1}^{L_k-1} \sum_{s'=0}^{s-1} a_t b_k \mathbb{I}[k = \kappa(t)] \mathbb{I}[s' = t - t_{\kappa(t)}] \\
&= \sum_{t=1}^{T} \sum_{s=1}^{L_{\kappa(t)}-1} \sum_{s'=0}^{s-1} a_t b_{\kappa(t)} \mathbb{I}[s' = t - t_{\kappa(t)}] \\
&= \sum_{t=1}^{T} \sum_{s=1}^{L_{\kappa(t)}-1} a_t b_{\kappa(t)} \mathbb{I}[s - 1 \geq t - t_{\kappa(t)}] \\
&= \sum_{t=1}^{T} a_t b_{\kappa(t)} \left( L_{\kappa(t)} - 1 - 1 - (t - t_{\kappa(t)}) + 1 \right) \mathbb{I}[L_{\kappa(t)} - 1 \geq t - t_{\kappa(t)}] \\
&= \sum_{t=1}^{T} a_t b_{\kappa(t)} \left[ L_{\kappa(t)} - (t - t_{\kappa(t)}) - 1 \right]_+ .
\end{aligned}
$$

$\square$

## D.6 PROOF OF LEMMA 2

*Proof.* Consider $k$-th cycle for item $i$ with cycle length of $L_k^i$. By definition, we have $\hat{y}_{t_k+s}^i < y_{t_k+s}^i$ for $s = 1, \ldots, L_k^i - 1$. By Lemma 3, $y_{t_k+s}^i = x_{t_k+s}^i > 0$. Therefore, we have

$$
\begin{aligned}
y_{t_k}^i &\geq x_{t_k+1}^i + d_{t_k}^i = y_{t_k+1}^i + d_{t_k}^i \\
&\geq \ldots \\
&\geq x_{t_k+L_k^i-1}^i + \sum_{s=0}^{L_k^i-2} d_{t_k+s}^i
\end{aligned}
$$

If $L_k^i > L_{\max}$, then $y_{t_k}^i > D$ because $x_{t_k+L_k^i-1}^i > 0$ and $\sum_{s=0}^{L_k^i-2} d_{t_k+s}^i \geq \sum_{s=0}^{L_{\max}-1} d_{t_k+s}^i \geq D$. This contradicts $y_{t_k}^i \leq D$. $\square$

## E PROOF OF THEOREM 2

*Proof.* We start by defining a set of the restart rounds as $t_1, \ldots, t_n, t_{n+1}$, where the $i$-th restart occurs at $t_i$ and we set $t_1 = 1$ and $t_{n+1} = T + 1$. We assign labels to the parameter used in each restart as $L_1, \ldots, L_n$, where $L_i = 2^{i-1}$. In our algorithm, the base learner in $t_i, \ldots t_{i+1}$ is $\mathcal{E}(2L_i, T)$. Note that since $L_n$ is at most $2L_{\max}$, we have $n \leq \log_2 L_{\max} + 2$. The regret can be

divided into:

$$
\sum_{t=1}^{T} \left( \langle g_t, \hat{y}_t - u_t \rangle + 2GL_t^* \|\hat{y}_t - \hat{y}_{t+1}\|_1 \right) = \sum_{i=1}^{n} \sum_{t=t_i}^{t_{i+1}-1} \left( \langle g_t, \hat{y}_t - u_t \rangle + 2GL_t^* \|\hat{y}_t - \hat{y}_{t+1}\|_1 \right)
$$

$$
= \sum_{i=1}^{n} \sum_{t=t_i}^{t_{i+1}-1} \left( \langle g_t, \hat{y}_t - u_t \rangle + 2GL_i \|\hat{y}_t - \hat{y}_{t+1}\|_1 \right) + \sum_{i=1}^{n} \sum_{t=t_i}^{t_{i+1}-1} 2G(L_t^* - L_i)\|\hat{y}_t - \hat{y}_{t+1}\|_1 .
$$

$$(21)$$

For the first term, using the assumptions for $\mathcal{R}_{L,T}^{\mathcal{E}(L,T)}$, we have

$$
\sum_{i=1}^{n} \sum_{t=t_i}^{t_{i+1}-1} \left( \langle g_t, \hat{y}_t - u_t \rangle + 2GL_i \|\hat{y}_t - \hat{y}_{t+1}\|_1 \right) \leq \sum_{i=1}^{n} \mathcal{R}_{2L_i,T}^{\mathcal{E}(2L_i,T)}
$$

$$
\leq \left( \sum_{i=1}^{n} 2^\alpha L_i^\alpha \right) \mathcal{R}(T)
$$

$$
\leq \left( \sum_{i=1}^{n} 2^{\alpha i} \right) \mathcal{R}(T)
$$

$$
\leq C(\alpha) L_{\max}^\alpha \mathcal{R}(T) ,
$$

where $C(\alpha)$ is an $\alpha$-dependent constant. For the first inequality, we use the fact that when an algorithm guarantees an upper bound $\mathcal{R}_{L,T}^{\mathcal{E}(L,T)}$ for regret $\tilde{R}_T(L)$, it also ensures that $\tilde{R}_{T'}(L) \leq \mathcal{R}_{L,T}^{\mathcal{E}(L,T)}$ for $T' \leq T$. This can be observed by setting $g_t = \mathbf{0}$ for $t \in \{T'+1, \dots, T\}$, which extends the series $\{g_t\}_{t=1}^{t=T'}$ in $\tilde{R}_{T'}(L)$ to $\{g_t\}_{t=1}^{t=T}$. This allows us to apply the same bound $\mathcal{R}_{L,T}^{\mathcal{E}(L,T)}$ to $\tilde{R}_{T'}(L)$.

In the second term of Eq. (21), positive contribution comes from the rounds where the parameter $L_i$ underestimates $L_t^*$: $L_t^* > L_i$. Suppose the parameter is set to $L_i$ and the algorithm observes that a cycle starts at round $t$. The algorithm can detect that the cycle length is longer than $L_i$ if it has not finished at $t + L_i - 1$. Therefore, the underestimated period is at most $L_i$. The second term is bounded as

$$
\sum_{i=1}^{n} \sum_{t=t_i}^{t_{i+1}-1} 2G(L_t^* - L_i)\|\hat{y}_t - \hat{y}_{t+1}\|_1 \leq 2GL_{\max} \sum_{i=1}^{n} \sum_{t=t_i}^{t_{i+1}-1} \|\hat{y}_t - \hat{y}_{t+1}\|_1 \mathbb{I}[L_t^* > L_i]
$$

$$
\leq C_2 G L_{\max} \sum_{i=1}^{n} \sum_{t=t_i}^{t_{i+1}-1} L_i^{-\beta} \mathbb{I}[L_t^* > L_i]
$$

$$
\leq C_2 G L_{\max} \sum_{i=1}^{n} L_i^{1-\beta}
$$

$$
= C_2 G L_{\max} \sum_{i=1}^{n} 2^{(i-1)(1-\beta)}
$$

$$
= \begin{cases} \mathcal{O}(L_{\max}^{2-\beta}) & (\beta < 1), \\ \mathcal{O}(L_{\max} \log L_{\max}) & (\beta = 1), \\ \mathcal{O}(L_{\max}) & (\beta > 1), \end{cases}
$$

where $C_2$ is a constant. Combining these two inequalities concludes the proof. $\qquad \square$

## F    PROOF OF THEOREM 3

Below, in order to match the standard expression, we introduce $D' := 2D$ which indicates the upper bound of the diameter of $\mathcal{C}(\mathbf{0})$:

$$
\|x - y\|_2 \leq \|x - y\|_1 \leq \|x\|_1 + \|y\|_1 \leq 2D =: D'
$$

for any $x, y \in \mathcal{C}(\mathbf{0})$.

*Proof.* We first bound $\tilde{R}_T(L)$. The first term of $\tilde{R}_T(L)$ is bounded by Lemma 8. For the second term, we have

$$GL \sum_{t=1}^{T} \|\hat{y}_t - \hat{y}_{t+1}\|_1 = GL \sum_{t=1}^{T} \|\hat{y}_t - (\Pi_{\mathcal{C}(\mathbf{0})}(\hat{y}_t - \eta g_t))\|_1$$

$$\leq GL\sqrt{N} \sum_{t=1}^{T} \|\hat{y}_t - (\Pi_{\mathcal{C}(\mathbf{0})}(\hat{y}_t - \eta g_t))\|_2$$

$$\leq \eta GL\sqrt{N} \sum_{t=1}^{T} \|g_t\|_2$$

$$\leq \eta\sqrt{N} G^2 LT.$$

Combining the first and second upper bounds, we have

$$\tilde{R}_T(L) \leq \frac{D'}{2\eta}(3D' + 2P_T) + \eta G^2 \left(\sqrt{N}L + \frac{1}{2}\right) T.$$

Then, by setting $\eta$ to

$$\eta = \frac{D'}{G}\sqrt{\frac{(3 + 2P_T/D')}{2(\sqrt{N}L + 1/2)T}},$$

we have

$$\tilde{R}_T(L) \leq G\sqrt{2D'(3D' + 2P_T)(\sqrt{N}L + 1/2)T}$$

$$\leq 2GD'N^{1/4}\sqrt{(3 + 2P_T/D')LT}$$

$$= \mathcal{O}(\sqrt{L(1 + P_T)T}).$$

Specifically, for $(3 + 2P_T/D')L \leq T$, $\|\hat{y}_t - \hat{y}_{t+1}\|_1 \leq \eta\sqrt{N}G \leq \mathcal{O}(L^{-1})$. This corresponds to $\alpha = 1/2$ and $\beta = 1$ in Theorem 2, which leads to

$$R_T \leq \mathcal{O}(\sqrt{L_{\max}(1 + P_T)T} + L_{\max}).$$

$\square$

**Lemma 8.** *Alg. 3 ensures*

$$\sum_{t=1}^{T} \langle g_t, \hat{y}_t - u_t \rangle \leq \frac{3D'^2}{2\eta} + \frac{D'}{\eta}P_T + \frac{\eta G^2 T}{2}.$$

*Proof.* Let us define $\hat{y}'_{t+1} := \hat{y}_t - \eta g_t$. For any $t \in [T]$, setting $u_{T+1} = 0$, we have

$$\langle g_t, \hat{y}_t - u_t \rangle = \frac{1}{\eta}\langle \hat{y}_t - \hat{y}'_{t+1}, \hat{y}_t - u_t \rangle$$

$$= \frac{1}{2\eta}\left(\|\hat{y}_t - u_t\|_2^2 + \|\hat{y}'_{t+1} - \hat{y}_t\|_2^2 - \|\hat{y}'_{t+1} - u_t\|_2^2\right)$$

$$\leq \frac{1}{2\eta}\left(\|\hat{y}_t - u_t\|_2^2 + \eta^2\|g_t\|_2^2 - \|\hat{y}_{t+1} - u_t\|_2^2\right)$$

$$= \frac{1}{2\eta}\left(\|\hat{y}_t - u_t\|_2^2 - \|\hat{y}_{t+1} - u_{t+1}\|_2^2 + \|\hat{y}_{t+1} - u_{t+1}\|_2^2 - \|\hat{y}_{t+1} - u_t\|_2^2 + \eta^2\|g_t\|_2^2\right)$$

$$= \frac{1}{2\eta}\left(\|\hat{y}_t - u_t\|_2^2 - \|\hat{y}_{t+1} - u_{t+1}\|_2^2 + \langle 2\hat{y}_{t+1} - u_{t+1} - u_t, u_t - u_{t+1}\rangle + \eta^2\|g_t\|_2^2\right)$$

$$\leq \frac{1}{2\eta}\left(\|\hat{y}_t - u_t\|_2^2 - \|\hat{y}_{t+1} - u_{t+1}\|_2^2 + 2D'\|u_t - u_{t+1}\|_1 + \eta^2\|g_t\|_2^2\right)$$

$$\leq \frac{1}{2\eta}\left(\|\hat{y}_t - u_t\|_2^2 - \|\hat{y}_{t+1} - u_{t+1}\|_2^2\right) + \frac{D'}{\eta}\|u_t - u_{t+1}\|_1 + \frac{\eta G^2}{2}.$$

In the third line, we use the inequality $\|\Pi_{\mathcal{C}(\mathbf{0})}(x) - \Pi_{\mathcal{C}(\mathbf{0})}(y)\|_2 \leq \|x - y\|_2$ for any $x, y \in [0, D]^N$. The summation over $t \in [T]$ leads to

$$\sum_{t=1}^{T} \langle g_t, \hat{y}_t - u_t \rangle \leq \frac{1}{2\eta} \|\hat{y}_1 - u_1\|_2^2 + \frac{D'}{\eta} \sum_{t=1}^{T} \|u_t - u_{t+1}\|_1 + \frac{\eta G^2 T}{2}$$

$$\leq \frac{3D'^2}{2\eta} + \frac{D'}{\eta} \sum_{t=2}^{T} \|u_{t-1} - u_t\|_1 + \frac{\eta G^2 T}{2}. \tag{22}$$

In the final line, we use $\|u_T - u_{T+1}\|_1 \leq D'$. □

## G  PROOF OF THEOREM 4

In this section, we abuse a notation, eliminating hats in the main paper: $v_{t+1}^k$ and $y_{t+1}^k$ are output of $\mathcal{A}^k$ and $\mathcal{B}^k$ in round $t$, respectively. $y_{t+1} = v_{t+1}^K$ describes the final output of Alg. 5. For the sake of brevity, we also define

$$\hat{\ell}_t(y^k) := \langle g_t, y_t^k \rangle + GL\|y_t^k - y_{t+1}^k\|_1,$$
$$\hat{\ell}_t(v^k) := \langle g_t, v_t^k \rangle + GL\|v_t^k - v_{t+1}^k\|_1.$$

Then, the bit for the combiner $k$ is defined as

$$b_t^k := \frac{\hat{\ell}_t(v^{k-1}) - \hat{\ell}_t(y^k)}{3GD'N^{1/4}\sqrt{L}}. \tag{23}$$

Recall that $v_t^k = (1 - p_t^k)v_t^{k-1} + p_t^k y_t^k$, where $p_t^k$ is the weight computed by the $k$-th combiner $\mathcal{B}^k$.

*Proof of Theorem 4.* From Lemma 9 and 16, we have $\mathcal{R}_{T,L}^{\mathcal{E}} = \mathcal{O}(\sqrt{L(1 + P_T)T\log T})$ and $\beta = 1$. Therefore, by Theorem 2, we obtain

$$R_T \leq \mathcal{O}(\sqrt{L_{\max}(1 + P_T)T\log T} + L_{\max}).$$

□

**Lemma 9.** *For $T \geq \max(\sqrt{L}\log_2 T, e)$, Alg. 5 ensures*

$$\tilde{R}_T(L) \leq \mathcal{O}(\sqrt{L(1 + P_T)T\log T}).$$

*Proof.* We first consider a large $P_T$ case, where the following inequality holds:

$$3 + \frac{2P_T}{D'} > \frac{T}{32L\log T}.$$

Then the regret is bounded as

$$\sum_{t=1}^{T} \langle g_t, y_t - u_t \rangle + GL\|y_t - y_{t+1}\|_1 = \sum_{t=1}^{T} \langle g_t, v_t^K - u_t \rangle + GL\|v_t^K - v_{t+1}^K\|_1$$

$$\leq 3GD'N^{1/4}T = 3GD'N^{1/4}\sqrt{T} \cdot \sqrt{T}$$

$$\leq 3GD'N^{1/4}\sqrt{32L(3 + 2P_T/D')T\log T}$$

$$\leq 24\sqrt{2}GDN^{1/4}\sqrt{L(3 + 2P_T/D)T\log T}$$

$$= \mathcal{O}(\sqrt{L(1 + P_T)T\log T}).$$

In the first inequality, we use Lemma 16 to bound the switching cost. Below, we consider the case of small $P_T$, where $3 + 2P_T/D' \leq T/(32L\log T)$. Theorem 3 shows that the optimal $\eta$ for OGD is given by $\eta^* = \alpha\sqrt{3 + 2P_T/D'}$, where $\alpha := D'/(G\sqrt{(2\sqrt{N}L + 1)T})$. On the other hand, we

define the learning rates of $\mathcal{A}^k$ as $\eta^k = \alpha\sqrt{2^{i-1}}$, for $k = 1, \ldots, K$, where $K = \lfloor\log_2\frac{T}{32L\log T}\rfloor + 1$. Because $K$ satisfies $3 + 2P_T/D' \le T/(32L\log T) \le 2^K$, there exists an $a \in [K]$ that satisfies

$$2^{a-1} \le 3 + \frac{2P_T}{D'} \le 2^a,$$

which implies $\eta^a \le \eta^* \le \sqrt{2}\eta^a$. Under $\eta^a$, the regret upper bound of OGD is given by

$$
\begin{aligned}
\sum_{t=1}^{T}\langle g_t, y_t^a - u_t\rangle + GL\|y_t^a - y_{t+1}^a\|_1 &\le \frac{D'^2}{2\eta^a}\left(3 + \frac{2P_T}{D'}\right) + \eta^a G^2\left(\sqrt{N}L + \frac{1}{2}\right)T \\
&\le \frac{\sqrt{2}D'^2}{2\eta^*}\left(3 + \frac{2P_T}{D'}\right) + \eta^* G^2\left(\sqrt{N}L + \frac{1}{2}\right)T \\
&\le \frac{\sqrt{2}+1}{\sqrt{2}}G\sqrt{D'(3D' + 2P_T)(\sqrt{N}L + 1/2)T} \\
&\le 3GD'N^{1/4}\sqrt{(3 + 2P_T/D')LT} \\
&\le 6GDN^{1/4}\sqrt{(3 + 2P_T/D)LT} \\
&= \mathcal{O}(\sqrt{L(1 + P_T)T}). \quad (24)
\end{aligned}
$$

Using such $a$, the dynamic regret with switching cost can be decomposed of

$$
\begin{aligned}
\tilde{R}_T(L) &= \sum_{t=1}^{T}\langle g_t, y_t - u_t\rangle + GL\|y_t - y_{t+1}\|_1 = \sum_{t=1}^{T}\langle g_t, v_t^K - u_t\rangle + GL\|v_t^K - v_{t+1}^K\|_1 \\
&= \sum_{t=1}^{T}\hat{\ell}_t(v^K) - \sum_{t=1}^{T}\langle g_t, u_t\rangle \\
&= \sum_{t=1}^{T}\left(\sum_{k=a+1}^{K}\hat{\ell}_t(v^k) - \hat{\ell}_t(v^{k-1})\right) + \sum_{t=1}^{T}(\hat{\ell}_t(v^a) - \hat{\ell}_t(y^a)) + \sum_{t=1}^{T}(\langle g_t, y_t^a - u_t\rangle + GL\|y_t^a - y_{t+1}^a\|_1).
\end{aligned}
$$

For the first term, we have

$$
\begin{aligned}
\sum_{t=1}^{T}\sum_{k=a+1}^{K}\hat{\ell}_t(v^k) - \hat{\ell}_t(v^{k-1}) &\stackrel{\text{Lemma 10}}{\le} -3GD'N^{1/4}\sqrt{L}\sum_{t=1}^{T}\sum_{k=a+1}^{K}\left(p_t b_t^k - \sqrt{L}|p_t - p_{t+1}|\right) \\
&\stackrel{\text{Lemmas 12 and 14}}{\le} 3GD'N^{1/4}\sqrt{L}\sum_{k=a+1}^{K}\left(U(n^k) + \frac{1}{\sqrt{L}} + 1\right) \\
&\stackrel{\text{Lemma 11}}{\le} 3GD'N^{1/4}\sqrt{L}\sum_{k=a+1}^{K}\left(4\sqrt{n^k\log T} + 2\right) \\
&\le 6GD'N^{1/4}\sqrt{L}\left(2\sqrt{n^a\log T}\sum_{k=1}^{K}\sqrt{2^{-k}} + (K - a)\right) \\
&\le 6GD'N^{1/4}\sqrt{L}\left(2(\sqrt{2}+1)\sqrt{n^a\log T} + 4\log T\right) \\
&\le 60GDN^{1/4}\sqrt{LT\log T} + 60GDN^{1/4}\sqrt{L}\log T.
\end{aligned}
$$

In the last line, we use $n^a = T2^{1-a} \leq T$. Similarly, for the second term, we have

$$\sum_{t=1}^{T} \hat{\ell}_t(v^a) - \hat{\ell}_t(y^a) \overset{\text{Lemma } 10}{\leq} -3GD'N^{1/4}\sqrt{L}\sum_{t=1}^{T}\left((p_t^a - 1)b_t^a - \sqrt{L}|p_t^a - p_{t+1}^a|\right)$$

$$\overset{\text{Lemmas } 12 \text{ and } 14}{\leq} 3GD'N^{1/4}\sqrt{L}\left(\frac{T}{n^a}\left(U(n^a) + \frac{2}{\sqrt{L}}\right) + U(n^a) + \frac{1}{\sqrt{L}} + 1\right)$$

$$\leq 3GD'N^{1/4}\sqrt{L}\left(\frac{T}{n^a}U(n^a) + U(n^a) + \frac{2T}{n^a} + 2\right)$$

$$\overset{\text{Lemma } 11}{\leq} 3GD'N^{1/4}\sqrt{L}\left(4\sqrt{T^2\log T/n^a} + 4\sqrt{n^a\log T} + \frac{2T}{n^a} + 2\right)$$

$$\leq 3GD'N^{1/4}\sqrt{L}\left(4\sqrt{(3 + 2P_T/D')T\log T} + 4\sqrt{T\log T} + 2(3 + 2P_T/D') + 2\right)$$

$$\leq 3GD'N^{1/4}\sqrt{L}\left(4\sqrt{(3 + 2P_T/D')T\log T} + 4\sqrt{T\log T} + 2\sqrt{6(3 + 2P_T/D')T} + 2\right)$$

$$\leq 3GD'N^{1/4}\sqrt{L}\left(9\sqrt{(3 + 2P_T/D')T\log T} + 6\sqrt{T\log T}\right)$$

$$\leq 54GDN^{1/4}\sqrt{(3 + 2P_T/D)LT\log T} + 36GDN^{1/4}\sqrt{LT\log T}.$$

For the fifth line, recall that $n^a/2 = T2^{-a} \leq T/(3 + 2P_T/D') \leq 2^{1-a} = n^a \leq T$. For the sixth line, because $P_T \leq TD'$, we use $3 + 2P_T/D' = 3 + 2\sqrt{P_T/D'}\cdot\sqrt{P_T/D'} \leq 3 + 2\sqrt{TP_T/D'} \leq \sqrt{2(9 + 4P_T/D')T} \leq \sqrt{6(3 + 2P_T/D')T}$.

Finally, the third term is bounded by Eq. (24), that is,

$$\sum_{t=1}^{T}\langle g_t, y_t^a - u_t\rangle + GL\|y_t^a - y_{t+1}^a\|_1 \leq 6GDN^{1/4}\sqrt{(1 + P_T/D)LT}.$$

Combining them, we have

$$\tilde{R}_T(L) \leq 60GDN^{1/4}\sqrt{L(3 + 2P_T/D)T\log T} + 96GDN^{1/4}\sqrt{LT\log T} + 60GDN^{1/4}\sqrt{L}\log T$$

$$= \mathcal{O}(\sqrt{L(1 + P_T)T\log T} + \sqrt{LT\log T} + \sqrt{L}\log T),$$

which finishes the proof. $\qquad\square$

**Lemma 10.**

$$\sum_{t=1}^{T}\left(\hat{\ell}_t(v^k) - \hat{\ell}_t(v^{k-1})\right) \leq -3GD'N^{1/4}\sqrt{L}\sum_{t=1}^{T}(p_t^k b_t^k - \sqrt{L}|p_t^k - p_{t+1}^k|),$$

$$\sum_{t=1}^{T}\left(\hat{\ell}_t(v^k) - \hat{\ell}_t(y^k)\right) \leq -3GD'N^{1/4}\sqrt{L}\sum_{t=1}^{T}((p_t^k - 1)b_t^k - \sqrt{L}|p_t^k - p_{t+1}^k|).$$

*Proof.* By using $v_t^k = (1 - p_t^k)v_t^{k-1} + p_t^k y_t^k$, we have

$$\hat{\ell}_t(v^k) = \langle g_t, v_t^k\rangle + GL\|v_t^k - v_{t+1}^k\|_1$$

$$= (1 - p_t^k)\langle g_t, v_t^{k-1}\rangle + p_t^k\langle g_t, y_t^k\rangle + GL\|(1 - p_t^k)v_t^{k-1} + p_t^k y_t^k - (1 - p_{t+1}^k)v_{t+1}^{k-1} - p_{t+1}^k y_{t+1}^k\|_1$$

$$= (1 - p_t^k)\langle g_t, v_t^{k-1}\rangle + p_t^k\langle g_t, y_t^k\rangle + (1 - p_t^k)GL\|(v_t^{k-1} - v_{t+1}^{k-1})\|_1 + p_t^k GL\|(y_t^k - y_{t+1}^k)\|_1$$

$$\qquad\qquad\qquad\qquad\qquad\qquad + GL\|(p_t^k - p_{t+1}^k)(y_{t+1}^k - v_{t+1}^{k-1})\|_1$$

$$\leq (1 - p_t^k)\hat{\ell}_t(v^{k-1}) + p_t^k\hat{\ell}_t(y^k) + GD'L|p_t^k - p_{t+1}^k|$$

Therefore, we have

$$
\begin{aligned}
\sum_{t=1}^{T}\left(\hat{\ell}_t(v^k) - \hat{\ell}_t(v^{k-1})\right) &\leq \sum_{t=1}^{T}\left(-p_t^k(\hat{\ell}_t(v^{k-1}) - \hat{\ell}_t(y^k)) + GD'L|p_t^k - p_{t+1}^k|\right) \\
&= \sum_{t=1}^{T}\left(-3GD'N^{1/4}\sqrt{L}p_t^k b_t^k + GD'L|p_t^k - p_{t+1}^k|\right) \\
&\leq -3GD'N^{1/4}\sqrt{L}\sum_{t=1}^{T}(p_t^k b_t^k - \sqrt{L}|p_t^k - p_{t+1}^k|),
\end{aligned}
$$

and

$$
\begin{aligned}
\sum_{t=1}^{T}\left(\hat{\ell}_t(v^k) - \hat{\ell}_t(y^k)\right) &\leq \sum_{t=1}^{T}\left((1 - p_t^k)(\hat{\ell}_t(v^{k-1}) - \hat{\ell}_t(y^k)) + GD'L|p_t^k - p_{t+1}^k|\right) \\
&\leq -3GD'N^{1/4}\sqrt{L}\sum_{t=1}^{T}((p_t^k - 1)b_t^k - \sqrt{L}|p_t^k - p_{t+1}^k|).
\end{aligned}
$$

$\square$

**Lemma 11** (Eq. (11) in Zhang et al. (2022a))**.** $U(n) \leq 4\sqrt{n\log T}$.

**Lemma 12** (The former part of Theorem 1 in Zhang et al. (2022a))**.** *Suppose* $T \geq e$ *and* $n^k \geq \max(8e, 16\log T)$*. For any bit sequence* $b_1^k, \ldots, b_T^k$ *such that* $|b_t^k| \leq 1/\sqrt{L} \leq 1$*, the following inequality holds under Alg.4:*

$$
-\sum_{t=1}^{T}(p_t^k b_t^k - \sqrt{L}|p_t^k - p_{t+1}^k|) \leq -\max\left(0, \sum_{t=1}^{T} b_t^k - \frac{T}{n^k}\left(U(n^k) + \frac{2}{\sqrt{L}}\right)\right) + U(n^k) + \frac{1}{\sqrt{L}} + 1
$$

Note that for $n^k = T2^{1-k}$ in our algorithm, $n^k \geq \max(8e, 16\log T)$ is satisfied because $n^k \geq n^K = T2^{1-K} \geq 32L\log T \geq 32L$. In the last inequality, we use $T \geq e$.

**Lemma 13** (The latter part of Theorem 1 in Zhang et al. (2022a))**.** *Under the setting in Lemma 12,*

$$
|p_t^k - p_{t+1}^k| \leq \frac{1}{\sqrt{L}}\left(\sqrt{\frac{1}{n^k}\log T} + \frac{1}{4T}\right).
$$

**Lemma 14.** *Assume* $T \geq \max(\sqrt{L}\log_2 T, e)$*. Then,* $|b_t^k| \leq 1/\sqrt{L}$*, for any* $k \in [K]$*.*

*Proof.*

$$
\begin{aligned}
|b_t^k| &= \frac{1}{3GD'N^{1/4}\sqrt{L}}\left|\hat{\ell}_t(v^{k-1}) - \hat{\ell}_t(y^k)\right| \\
&= \frac{1}{3GD'N^{1/4}\sqrt{L}}\left|\langle g_t, v_t^{k-1} - y_t^k\rangle + GL\|v_t^{k-1} - v_{t+1}^{k-1}\|_1 - GL\|y_t^k - y_{t+1}^k\|_1\right| \\
&\leq \frac{1}{3GD'N^{1/4}\sqrt{L}}\left(GD' + GL\max(\|v_t^{k-1} - v_{t+1}^{k-1}\|_1, \|y_t^k - y_{t+1}^k\|_1)\right) \\
&\leq \frac{1}{\sqrt{L}}.
\end{aligned}
$$

In the last line, we use Lemmas 15 and 16. $\square$

**Lemma 15.** $\|y_t^k - y_{t+1}^k\|_1 \leq D'N^{1/4}/L$ *for any* $k \in [K]$*.*

*Proof.* Since $y_{t+1}^k$ is updated by OGD with the learning rate of $\eta^k$, we have

$$\|y_t^k - y_{t+1}^k\|_1 \le \eta^k \sqrt{N} \|g_t\|_2 \le \sqrt{N} G \cdot \frac{D'}{G} \sqrt{\frac{2^{k-1}}{(2N^{1/2}L+1)T}}$$

$$\le \frac{D'N^{1/4}}{\sqrt{L}} \sqrt{\frac{2^{K-1}}{2T}} \le \frac{D'N^{1/4}}{\sqrt{L}} \sqrt{\frac{1}{32L\log T}} \le \frac{D'N^{1/4}}{L},$$

which concludes the proof. $\qquad\square$

**Lemma 16.** *Assume $T \ge \max(\sqrt{L}\log_2 T, e)$. Then, $\|v_t^k - v_{t+1}^k\|_1 \le 2D'N^{1/4}/L$ for any $k \in [K]$.*

*Proof.* We show it by induction. For $k = 1$, since $v_t^1 = y_t^1$, the inequality holds by Lemma 15. Suppose the inequality

$$\|v_t^k - v_{t+1}^k\|_1 \le \frac{D'N^{1/4}}{L} + \frac{D'}{\sqrt{L}} \sum_{i=2}^k \left( \sqrt{\frac{1}{n^i}\log T} + \frac{1}{4T} \right), \tag{25}$$

holds for $k \ge 1$. We note that this inequality satisfies $LG\|v_t^k - v_{t+1}^k\|_1 \le 2GD'$ because

$$\|v_t^k - v_{t+1}^k\|_1 \le \frac{D'N^{1/4}}{L} + \frac{D'}{\sqrt{L}} \sum_{i=2}^k \left( \sqrt{\frac{1}{n^i}\log T} + \frac{1}{4T} \right)$$

$$\le \frac{D'N^{1/4}}{L} + \frac{D'}{\sqrt{L}} \sum_{i=2}^K \left( \sqrt{\frac{2^i}{2T}\log T} + \frac{1}{4T} \right)$$

$$\le \frac{D'N^{1/4}}{L} + \frac{D'}{\sqrt{L}} \left( \frac{2}{\sqrt{2}-1} \sqrt{\frac{\log T}{2T}} \sqrt{\frac{T}{32L\log T}} + \frac{\log_2 T}{4T} \right)$$

$$\le \frac{D'N^{1/4}}{L} + \frac{D'}{L} \left( \frac{(\sqrt{2}+1)}{4} + \frac{\sqrt{L}\log_2 T}{4T} \right)$$

$$\le \frac{D'N^{1/4}}{L} + \frac{D'}{L} \left( \frac{(\sqrt{2}+1)}{4} + \frac{1}{4} \right)$$

$$\le \frac{2D'N^{1/4}}{L}.$$

In the fifth line, we use $\sqrt{L}\log_2 T \le T$. We also note that the assumption $T \ge \max(\sqrt{L}\log_2 T, e)$ and Lemma 15 leads to $|b_t^{k+1}| \le 1/\sqrt{L}$ and Lemma 13 holds for $k+1$. Therefore, for $k+1$, we have

$$\|v_t^{k+1} - v_{t+1}^{k+1}\|_1 = \|(1-p_t^{k+1})v_t^k + p_t^{k+1}y_t^{k+1} - (1-p_{t+1}^{k+1})v_{t+1}^k - p_{t+1}^{k+1}y_{t+1}^{k+1}\|_1$$

$$\le \|(1-p_t^{k+1})(v_t^k - v_{t+1}^k) + p_t^{k+1}(y_t^{k+1} - y_{t+1}^{k+1}) - (p_t^{k+1} - p_{t+1}^{k+1})(v_{t+1}^k - y_{t+1}^{k+1})\|_1$$

$$\le (1-p_t^{k+1})\|v_t^k - v_{t+1}^k\|_1 + p_t^{k+1}\|y_t^{k+1} - y_{t+1}^{k+1}\|_1 + |p_t^{k+1} - p_{t+1}^{k+1}|\|v_{t+1}^k - y_{t+1}^{k+1}\|_1$$

$$\le (1-p_t^{k+1}) \left( \frac{D'N^{1/4}}{L} + \frac{D'}{\sqrt{L}} \sum_{i=2}^k \left( \sqrt{\frac{1}{n^i}\log T} + \frac{1}{4T} \right) \right) + p_t^{k+1}\frac{D'N^{1/4}}{L} + |p_t^{k+1} - p_{t+1}^{k+1}|D'$$

$$\le \frac{D'N^{1/4}}{L} + \frac{D'}{\sqrt{L}} \sum_{i=2}^k \left( \sqrt{\frac{1}{n^i}\log T} + \frac{1}{4T} \right) + |p_t^{k+1} - p_{t+1}^{k+1}|D'.$$

$$\overset{\text{Lemma 13}}{\le} \frac{D'N^{1/4}}{L} + \frac{D'}{\sqrt{L}} \sum_{i=2}^{k+1} \left( \sqrt{\frac{1}{n^i}\log T} + \frac{1}{4T} \right).$$

In the fourth line, we use Eq. (25) and Lemma 15. Here we observe that Eq. (25) holds for $k+1$. Hence, by induction, we conclude the proof. $\qquad\square$

# H   PROOF OF THEOREM 5

*Proof.* Let $D_g$ be a distribution of loss sequences, and $\mathcal{G}$ be the support of $D_g$. Then, we have

$$\mathbb{E}_{D_g}\left[\sum_{t\in[T]}\langle g_t, y_t - u\rangle\right] \le \sup_{\{g_t\}_t\in\mathcal{G}}\sum_{t\in[T]}\langle g_t, y_t - u\rangle.$$

Thus, we will obtain our lower bound by showing a lower bound of the expected regret. Moreover, we will construct a common distribution of instances for all algorithms. Hence, we can assume that the given algorithm is deterministic without loss of generality.

We can assume that $L_{\max} = 2L+1$ and $T = L_{\max}K$ for some $L, K > 0$ without loss of generality. Note that $L = \Theta(L_{\max})$. We divide $T$ rounds into $K$ cycles, where a cycle has $L_{\max}$ rounds. Let $t_k$ be the first round in the $k$-th cycle.

We fix $k \in [K]$ arbitrarily. We consider the following distribution of instances.

$$x_{t+1}^i = \begin{cases} y_t^i & t \in [t_k, t_k + 2L - 1] \\ 0 & t = t_k + 2L \end{cases} \quad \text{and}$$

$$g_t^i = \begin{cases} -\frac{G}{2} & \text{if } i = 1 \text{ and } t \in [t_k, t_k + L - 1] \\ \frac{G(\epsilon_k+1)}{2} & \text{if } i = 1 \text{ and } t \in [t_k + L, t_k + 2L - 1] \,, \\ 0 & \text{otherwise} \end{cases}$$

where $\epsilon_k$ is a Rademacher random variable, i.e., $\mathbb{P}(\epsilon_k = 1) = \mathbb{P}(\epsilon_k = -1) = \frac{1}{2}$. Note that the demands of items in these instances do not rely on given algorithm. Indeed, we have

$$\tilde{d}_t^i = \begin{cases} 0 & t \in [t_k, t_k + 2L - 1] \\ D & t = t_k + 2L \end{cases}$$

and $x_{t+1}^i = \max(0, y_t^i - \tilde{d}_t^i)$ for all $i \in [N]$ and $t \in [T]$. Note also that $L_{\max}$ is an upper bound of the sell-out period since $x_t^i$ becomes zero at the end of each cycle for all $i \in [N]$.

Then, we discuss the cumulative loss by an algorithm. We have

$$\sum_{t=t_k}^{t_k+2L}\langle g_t, y_t\rangle = \sum_{t=t_k}^{t_k+L-1}\langle g_t, y_t\rangle + \sum_{t=t_k+L}^{t_k+2L-1}\langle g_t, y_t\rangle$$

$$= \sum_{t=t_k}^{t_k+L-1} -\frac{G}{2}y_t^1 + \sum_{t=t_k+L}^{t_k+2L-1}\frac{G(\epsilon_k+1)}{2}y_t^1$$

$$\ge -\frac{GL}{2}y_{t_k+L-1}^1 + \sum_{t=t_k+L}^{t_k+2L-1}\frac{G(\epsilon_k+1)}{2}y_t^1,$$

where the inequality holds due to the definition of $x_t^i$ in the instances. Now, we focus on the second term on the right-hand side. Since $y_t^1 \ge y_{t_k+L-1}^1$ for all $t \in [t_k + L, t_k + 2L - 1]$, if $\epsilon_k = 1$, we have

$$\sum_{t=t_k+L}^{t_k+2L-1}\frac{G(\epsilon_k+1)}{2}y_t^1 \ge GLy_{t_k+L-1}^1.$$

On the other hand, if $\epsilon_k = -1$, we have

$$\sum_{t=t_k+L}^{t_k+2L-1}\frac{G(\epsilon_k+1)}{2}y_t^1 = 0.$$

Therefore, we obtain

$$\mathbb{E}\left[\sum_{t=t_k}^{t_k+2L}\langle g_t, y_t\rangle\right] \ge \mathbb{E}\left[-\frac{GL}{2}y_{t_k+L-1}^1 + \sum_{t=t_k+L}^{t_k+2L-1}\frac{G(\epsilon_k+1)}{2}y_t^1\right] \ge 0. \tag{26}$$

Next, we consider the cumulative loss by the comparator. Let $T' = LK$, $e_i \in \mathbb{R}^N$ be the $i$-th canonical vector, and $\mathcal{U} = \{\mathbf{0}, De_1\}$. Then, we have

$$
\begin{aligned}
\min_{u \in \mathcal{C}(\mathbf{0})} \sum_{t \in [T]} \langle g_t, u \rangle &= \min_{u \in \mathcal{C}(\mathbf{0})} \sum_{k \in [K]} \sum_{t=t_k}^{t_k+2L} \langle g_t, u \rangle \\
&\leq \min_{u \in \mathcal{U}} \sum_{k \in [K]} \sum_{t=t_k}^{t_k+2L} \langle g_t, u \rangle \\
&= \min_{u \in \mathcal{U}} \sum_{k \in [K]} \left( \sum_{t=t_k}^{t_k+L-1} -\frac{G}{2} u^1 + \sum_{t=t_k+L}^{t_k+2L-1} \frac{G(\epsilon_k+1)}{2} u^1 \right) \\
&= \min_{u \in \mathcal{U}} \sum_{k \in [K]} \frac{GDL\epsilon_k}{2} u^1 \\
&= \frac{GDL}{2} \min_{u' \in \{0,1\}} \sum_{k \in [K]} \epsilon_k u'. \quad (27)
\end{aligned}
$$

Combining (26) and (27), we obtain

$$
\begin{aligned}
\mathbb{E}\left[ \sum_{t \in [T]} \langle g_t, y_t \rangle - \min_{u \in \mathcal{C}(\mathbf{0})} \sum_{t \in [T]} \langle g_t, u \rangle \right] &= \mathbb{E}\left[ \sum_{k \in [K]} \sum_{t=t_k}^{t_k+2L} \langle g_t, y_t \rangle - \min_{u \in \mathcal{C}(\mathbf{0})} \sum_{k \in [K]} \sum_{t=t_k}^{t_k+2L} \langle g_t, u \rangle \right] \\
&\geq -\frac{GDL}{2} \mathbb{E}\left[ \min_{u' \in \{0,1\}} \sum_{k \in [K]} \epsilon_k u' \right] \\
&= \frac{GDL}{2} \mathbb{E}\left[ \max_{u' \in \{0,1\}} \sum_{k \in [K]} \epsilon_k u' \right],
\end{aligned}
$$

where the last equality is derived from the fact that $-\epsilon_k$ is a Rademacher random variable. Finally, we obtain

$$
\begin{aligned}
\frac{GDL}{2} \mathbb{E}\left[ \max_{u' \in \{0,1\}} \sum_{k \in [K]} \epsilon_k u' \right] &= \frac{GDL}{4} \mathbb{E}\left[ \left| \sum_{k \in [K]} \epsilon_k \right| \right] \\
&\geq \frac{GDL}{4} \sqrt{K} \geq \Omega(GD\sqrt{L_{\max} T}),
\end{aligned}
$$

where we used $\max(a, b) = \frac{a+b}{2} + \frac{|a-b|}{2}$ in the equality, Khintchine inequality in the second inequality, and $K = \Theta(T/L)$ in the last inequality. $\qquad \square$

# I   EXPERIMENTS

We present the results of numerical experiments using synthetic demand data. We conduct experiments varying the value of $T \in [2000, 5000, 10000, 20000, 50000]$ and measure the regret for each algorithm. We consider an inventory system for a single item with a warehouse capacity of $D = 1$, and a Newsvendor loss of $\ell_t(y) = 5\max(d_t - y, 0) + \max(y - d_t, 0)$, where $d_t$ is the demand of round $t$. The demands are artificially generated as $d_t = D/2(1 + (1 - \epsilon(T))\sin(w(T)t))$, where $w(T) = 2\pi \log T / T$ and $\epsilon(T) = 1/\log T$. This parameterization ensures $L_{\max} \sim \mathcal{O}(\log T)$ and demand fluctuation $\sum_{t=2}^{T} |d_t - d_{t-1}| \sim \mathcal{O}(\log T)$, which are dominated by $\epsilon(T)$, and $w(T)$, respectively. We adopt the ideal comparator $u_t = d_t$, which incurs zero loss and gives $P_T = \sum_{t=2}^{T} |d_t - d_{t-1}|$. Initial inventory level and initial order are set to zero and $1/2$, respectively. We set the parameter $\gamma$ for MaxCOSD as $\gamma = 0.5\rho/D$ where $\rho$ represents the minimum of the demand series (note that we consider a deterministic demand in this experiment). We note that OGD requires $P_T$ as an input, whereas SOGD does not.

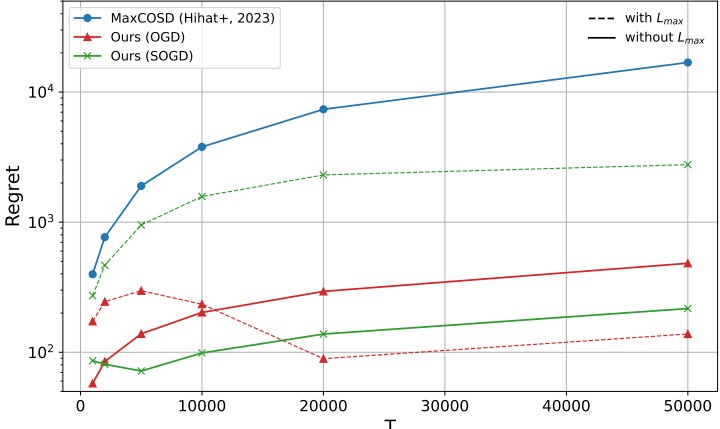

Figure 1: Experimental Results

The results are shown in Fig. 1. In the experiment, our algorithms significantly outperform the baseline (MaxCOSD). We observe that the algorithms using the doubling trick (solid lines) sometimes achieve lower regret than those with $L_{\max}$ information (dashed lines). This is because when using $L_{\max}$, the learning rate is set smaller than that used in the doubling trick case. As a result, it requires longer time to shift from the initial value to an appropriate order level, which can deteriorate the performance.

## J  THE USE OF LARGE LANGUAGE MODELS

In this paper, we used large language models to refine and check our writing; we did not use them for any other significant tasks.

