# OpenReview forum: "Online Inventory Optimization in Non-Stationary Environment"
_ICLR.cc/2026/Conference — ICLR 2026 Poster_

### Official Review · Reviewer_xAJn · 2025-11-01

**Soundness:** 2
**Presentation:** 2
**Contribution:** 3
**Rating:** 4
**Confidence:** 2

**Summary:**

This paper studies the problem of online inventory optimization, which is a setting where the decision-maker sequentially chooses the order-up-to level such that the warehouse capacity is not exceeded, with the goal to minimize the total cost according to some loss function (that is revealed after choosing an action). They use the "sell-out period" as a measure of the difficulty of the environment. They propose an algorithm that uses the doubling trick to handle the fact that the sell-out period is not known, a base smoothed online convex optimization algorithm, and also a projection from the output of the base algorithm to the feasible set.

**Strengths:**

- The algorithm appears to improve the regret rate beyond prior work that studied this problem. However, it is not clear that the referenced works are using the same assumptions as this paper.
- The algorithm appears to provide dynamic regret rates in this setting, which appear to be new. There is a clear motivation for the utility of such analysis.
- The use of smoothed OCO in this problem is a nice idea, although it is unclear to me how novel this idea is with respect to the literature.

**Weaknesses:**

- The comparison in Table 1 does not appear to be quite a fair comparison. In particular, the warehouse constraint in prior work Hihat et al. (2023) is a general convex set, whereas the one in the paper restricts the sum of the items to be less than the capacity. This seems to be quite a restrictive assumption of the items take up different space in the warehouse.
- I found some of the presentation to be poor to the point that it was difficult to verify the theoretical results in the paper. I point these out and ask for clarification in the Questions section.

**Questions:**

- In Remark 1, the paper seems to suggest that the gradient of $\ell(y_t) = p \max(d_t - y_t)$ can be computed without knowledge of $d_t$. However, this is not correct.
- It seems that in the definition of $L_\max$, the sum over the items of the demand is greater than capacity. Otherwise, it seems to not make sense.
- I found the first sentence of Definition 2 to be nonsensical.
- Do all of the works in Table 1 use the same notion of $L_\max$? I understand that the notation is different, but is the underlying quantity the same?

---

> ### Author Response · Authors · 2025-11-21
>
> Thank you for your insightful comments on our paper.
>
> > The comparison in Table 1 does not appear to be quite a fair comparison. In particular, the warehouse constraint in prior work Hihat et al. (2023) is a general convex set, whereas the one in the paper restricts the sum of the items to be less than the capacity. This seems to be quite a restrictive assumption of the items take up different space in the warehouse.
>
> We acknowledge that the difference in the capacity constraint is not addressed in Table 1 and we will revise it accordingly.
> Meanwhile, as we emphasized in Remark 1, no existing work has established theoretically guaranteed algorithms for dynamic environments, even under the linear constraint. We consider that filling this gap is a significant contribution.
> In addition, we note that our linear constraint is applicable to cases in which each item occupies a different amount of space.
> Specifically, the weighted-sum capacity constraint, $\sum_i a^i y^i_t \leq D$ where $a^i > 0$ is reduced to our linear constraint by redefining $a^i y^i_t \rightarrow y^i_t$ and $a^i d^i_t \rightarrow d^i_t$.
> Although this is still a special case of the general convex set, we believe this extension covers a broad range of practical cases.
>
> > In Remark 1, the paper seems to suggest that the gradient of $ l_t$ can be computed without knowledge of $d_t$. However, this is not correct.
>
> In Remark 1, our intention is to state that the subgradient can be obtained without knowing the demand quantity, only by observing whether a stockout occurred.
> We acknowledge that this point is unclear and we will clarify it in the revised version.
>
> > It seems that in the definition of $L_{\max}$, the sum over the items of the demand is greater than capacity. Otherwise, it seems to not make sense.
>
> We acknowledge that the first sentence in Definition 1 may be misleading, and will revise it.
> We note that the definition based on the demand sum and warehouse capacity is insufficient because the scale of each item's demand cannot be fully constrained.
> In our definition, $L_{\max}$ ensures that, for any item $i$ and any stock level, the zero-order strategy during $L_{\max}$ rounds reduces its stock level to zero.
> In contrast, the definition using demand sum does not always provide this guarantee.
> For example, suppose we have two items with demands $d^1_t = D/2$ and $d^2_t = 0$, respectively.
> The definition using demand sum leads to $L_{\max} = 2$.
> However, since $d^2_t = 0$, the zero-order strategy for item $2$ never reduces its stock level.
>
> > I found the first sentence of Definition 2 to be nonsensical.
>
> We have found that our current definition of $\mathcal{S}_i$ is self-referential and does not reflect our intention.
> $\mathcal{S}_i$ denotes the set of rounds $t \in [T]$ that satisfy $y^i_t \leq \hat{y}^i_t$.
> We will correct it in the revised version.
>
> > Do all of the works in Table 1 use the same notion of $L_{\max}$? I understand that the notation is different, but is the underlying quantity the same?
>
> Although most papers do not directly define $L_{\max}$, the leading-order dependence of their regret on $L_{\max}$ can be inferred from the parameters describing the demand characteristics. We listed the mapping to $L_{\max}$ in footnote 1.

---

### Official Review · Reviewer_KfMr · 2025-11-02

**Soundness:** 3
**Presentation:** 3
**Contribution:** 3
**Rating:** 6
**Confidence:** 2

**Summary:**

This paper addresses Online Inventory Optimization (OIO) in non-stationary environments, a variant of Online Convex Optimization (OCO). It proposes a two-stage projection algorithm (base learner + feasible-region projection) that connects OIO to Smoothed OCO (SOCO), achieving near-optimal dynamic regret $\tilde{O}(\sqrt{L_{max}T(1+P_T)})$. The paper also provides the first $\Omega(\sqrt{L_{max}T})$ lower bound for OIO.

**Strengths:**

1. The two-stage projection strategy effectively eliminates carryover stock constraints by linking OIO to SOCO, a key innovation addressing dynamic environment limitations of prior OIO works.
2. The paper establishes matching upper/lower bounds for OIO regret, and derives a lower bound as a valuable byproduct.

**Weaknesses:**

1. The paper does not evaluate the computational overhead of its algorithm. For example, the SOGD base learner requires $K=\lfloor\log_2 T/(32\max(L,1)\log T)\rfloor+1$ experts and combiners, and the doubling trick involves restarting the base learner; the growth of computational cost with T or $L_{max}$ is unreported, which is critical for real-time inventory applications.

2. The ideal comparator $u_t=d_t$ assumes $d_t\leq D$ to satisfy $u_t\in C(0)$. However, the paper does not discuss how the algorithm performs when demand $d_t>D$, which is a common real-world scenario, nor does it clarify whether the regret bound remains valid when $u_t$ must be capped at D.

**Questions:**

1. Could you provide preliminary insights into extending the framework to include lead time or fixed ordering costs?
2. In the high-probability extension of $L_{max}$, can you quantify how $\delta$ influences the choice of $L_{max}$?

---

> ### Author Response · Authors · 2025-11-21
>
> Thank you for your valuable comments on our paper.
>
> > The paper does not evaluate the computational overhead of its algorithm. For example, the SOGD base learner requires K experts and combiners, and the doubling trick involves restarting the base learner; the growth of computational cost with $T$ or $L_{\max}$ is unreported, which is critical for real-time inventory applications.
>
> As the reviewer points out, in order to suppress the dynamic regret upper bound, we adopt a meta-algorithm that incurs a computational cost of $\mathcal{O}(KT) = \tilde{\mathcal{O}}(T \log T)$. We note that such overhead is common in the OCO or SOCO setting, when using meta-algorithms for non-stationary environments. We will mention this computational overhead in the revised manuscript.
>
> > The ideal comparator $u_t = d_t$ assumes $d_t \leq D$ to satisfy $u_t \in \mathcal{C}(0)$. However, the paper does not discuss how the algorithm performs when demand $d_t > D$, which is a common real-world scenario, nor does it clarify whether the regret bound remains valid when $u_t$ must be capped at $D$.
>
> Our algorithm uses only the subgradient for updates; therefore, both our algorithm and analysis are applicable whenever the subgradient for $d_t > D$ is well-defined and observable. For example, in the Newsvendor loss, this subgradient is the same as the one observed when a stockout occurs.
>
> > Could you provide preliminary insights into extending the framework to include lead time or fixed ordering costs?
>
> For OIO with lead time, we expect that theoretically guaranteed algorithms can be obtained by combining our framework with the delayed feedback setting. In contrast, OIO with fixed ordering costs is more challenging because the ordering cost is nonconvex, which causes OCO algorithms to lose their theoretical guarantees. We leave these extensions for future work.
>
> > In the high-probability extension of $L_{\max}$, can you quantify how $\delta$ influences the choice of $L_{\max}$?
>
> We provided an analysis of the high-probability extension of $L_{\max}$ in Proposition 1 in Appendix and showed that, under the condition stated in Proposition 1, the  $\delta$-dependent term appears as $\log (N T/\delta)$.

---

### Official Review · Reviewer_KX9o · 2025-11-03

**Soundness:** 3
**Presentation:** 3
**Contribution:** 3
**Rating:** 6
**Confidence:** 3

**Summary:**

This paper addresses the Online Inventory Optimization (OIO) problem , which is an extension of Online Convex Optimization (OCO) that includes constraints from carryover stock. The authors make a strong case that the standard static regret metric is unsuitable for OIO, especially in non-stationary environments where demand fluctuates, as the best fixed strategy is often a poor comparator.  The main contribution is a new algorithm that provides a near-optimal dynamic regret guarantee , comparing the algorithm's performance to a time-varying sequence of decisions. This approach cleverly reveals a connection between OIO and Smoothed Online Convex Optimization (SOCO). This connection allows the authors to develop an algorithm with a dynamic regret bound of $\tilde{\mathcal{O}}(\sqrt{L_{max}T(1+P_{T})})$ and an improved static regret of $\tilde{\mathcal{O}}(\sqrt{L_{max}T})$, which is a $\sqrt{L_{max}}$ improvement over existing work. The authors also provide a matching $\Omega(\sqrt{L_{max}T})$ lower bound for the static regret, resolving an open question from prior literature.

**Strengths:**

1. The paper is well structured and self-contained, with helpful summaries of prior results (Table 1). The example provided on page 1 does a good job of illustrating why dynamic regret is the more appropriate metric.

2. The connection between OIO and SOCO via the projection lemma (Lemma 1) is interesting. It reduces a stateful inventory constraint problem to a well-studied smoothed OCO form.

3. The paper provides the first near-optimal dynamic regret bound for the setting. Furthermore, it improves the static regret bound and provides a matching lower bound, which is solid. The appearance of $\sqrt{L_{\max}}$ is well motivated.

**Weaknesses:**

1. The paper is purely theoretical and lack of experimental validation.

2. The paper simplifies the problem by assuming a linear warehouse capacity constraint.  The authors admit this is a limitation and that their proofs rely on it. This is a fair limitation, but it does reduce the generality of the result.

3. The connection to adaptive regret or meta-OCO methods (e.g., MetaGrad, Ader) is acknowledged, but it’s unclear if simpler baselines could reach similar dynamic regret under relaxed constraints.

**Questions:**

1. Could you provide a bit more intuition for $L_{\max}$? For example, in a simple stochastic setting with i.i.d. demand,  what would $L_{max}$ correspond to?

2. The doubling trick restarts the base learner and the set $\mathcal{L}_t$ seems to include the lengths of all previously completed cycles plus the current lengths of all active cycles. Does tracking this set introduce any significant computational or memory overhead, especially for large $N$?

---

> ### Author Response · Authors · 2025-11-21
>
> Thank you for your constructive comments on our paper.
>
> > The paper is purely theoretical and lack of experimental validation.
>
> We conducted numerical experiments using synthetic demand data and provided the results in Appendix I. In our experiments, we observed that our algorithms significantly outperform the baseline (Hihat et al., 2023). We also compared the performance with and without knowledge of $L_{\max}$, as well as adopting OGD and SOGD as the base learners. While we acknowledge that we have not conducted experiments on real data, we hope that this section addresses the reviewer's concern.
>
>
> > The connection to adaptive regret or meta-OCO methods (e.g., MetaGrad, Ader) is acknowledged, but it’s unclear if simpler baselines could reach similar dynamic regret under relaxed constraints.
>
> As the reviewer points out, we acknowledge that, in the standard OCO setting, the OGD algorithm can achieve $\mathcal{O}(\sqrt{(1 + P_T)T})$-dynamic regret upper bound if the learner knows $P_T$ in advance. However, as mentioned in the sentences below Theorem 3, this parameter is difficult to set a priori, which motivates the use of a meta-algorithm.
> We will add this point to the revised version.
>
> > Could you provide a bit more intuition for $L_{\max}$? For example, in a simple stochastic setting with i.i.d. demand, what would $L_{\max}$ correspond to?
>
> Informally speaking, $L_{\max}$ indicates that each item sells at least $D$ units for any interval $[t, t + L_{\max}]$. For a static demand $d$,  $L_{\max}$ is given by $\lceil D/d \rceil$. For a stochastic demand, $L_{\max}$ is defined as a high probability upper bound, as described in Remark 3. Specifically, Proposition 1 in Appendix states that if $\mathbb{P}\left[ d_t \geq \rho \right] \geq \mu$, then $L_{\max}$ is given by $\frac{2D}{\rho\mu} + \frac{3\log \frac{N T}{\delta}}{\mu^2}$. For example, if the i.i.d. demand satisfies $d_t \in \{0, D/2\}$ with $\mathbb{P}(d_t \geq \rho =  D/2) = \mu = 1/2$, the zero-order strategy for $4$ rounds (plus $\delta$-dependent additional rounds) leads to stockouts with high probability.
>
> > The doubling trick restarts the base learner and the set $L_t$ seems to include the lengths of all previously completed cycles plus the current lengths of all active cycles. Does tracking this set introduce any significant computational or memory overhead, especially for large $N$?
>
> This tracking requires only $\mathcal{O}(N)$ memory because the algorithm requires only the maximum length of previously completed cycles and the current lengths of all active cycles. We will clarify this point in the revised version.

---

### Public Comment · ~Koji_Ichikawa1 · 2026-03-02
**Camera-ready Update**

In the camera-ready version, in addition to incorporating the reviewers' feedback and fixing typos, we revised the proof of Theorem 2, which changes the order of the second term in its upper bound. Consequently, the second (non-leading) terms in Theorems 3 and 4 gain an additional logarithmic factor. The main conclusions of the paper remain unchanged.

More precisely, in Theorem 2 we updated the second term in the upper bound to a $\beta$-dependent order. As a result, the corresponding terms in Theorems 3 and 4 change from $\mathcal{O}(L_{\max})$ to $\mathcal{O}(L_{\max}\log L_{\max})$. These terms remain subdominant for a broad range of horizons, e.g., $T > L_{\max}\log^2 L_{\max}$.

---

### Meta-Review · Area_Chair_bHjq · 2026-01-07

**Summary:**

- This paper proposes a novel algorithm with regret bounds for online inventory optimization in. nonstationary environments.

- Three reviews were collected, with scores 6, 6, 4.

- The reviewers appreciate the theoretical contributions of regret upper bound and lower bound.

- Reviewer xAJn (score 4) finds the paper difficult to read and asks several clarifying questions, which are adddressed during the rebuttal.

**Reviewer Concerns:**

- Reviewers raise concerns on the comparison with the existing literature and the rebuttal addressed it successfully.

- The clarifying questions on the technical details were addressed by the rebuttal.

- Reviewer KfMr asks about the assumption of $d_t\leq D$. The rebuttal clarifies that the algorithm and the theoretical results both hold for the $d_t>D$ case.

- Reviewer xAJn pointed out the difference in the constraints considered in this paper and the prior work Hihat et al. 2023. The rebuttal acknowledged this limitation. This concern deserves more work in the future.

**Reviewer Scores:**

- Three reviews were collected, with scores 6, 6, 4.

- The reviewers with score 6 are not likely to revise their score.

- The reviewer with score 4 is likely to increase their score.

---

> ### Public Comment · ~Koji_Ichikawa1 · 2026-03-02
>
> Dear Area Chair,
>
> We sincerely thank you for taking the time to read our submission, reviews, and  comments, and for providing a thoughtful meta review.
> In the camera-ready version, in addition to incorporating the reviewers' feedback and fixing typos, we revised the proof of Theorem 2, which changes the order of the second term in its upper bound. Consequently, the second (non-leading) terms in Theorems 3 and 4 gain an additional logarithmic factor. The main conclusions of the paper remain unchanged.
> Details are summarized in our public comment and reflected in the revised manuscript.

---

### Decision · Program_Chairs · 2026-01-26

Accept (Poster)